# Follow-Your-Preference: Towards Preference-Aligned Image Inpainting

**Yutao Shen**[1][*] **Junkun Yuan**[2][*][†] **Toru Aonishi**[1] **Hideki Nakayama**[1] **Yue Ma**[3][†]
[1]The University of Tokyo [2]Zhejiang University [3]Tsinghua University

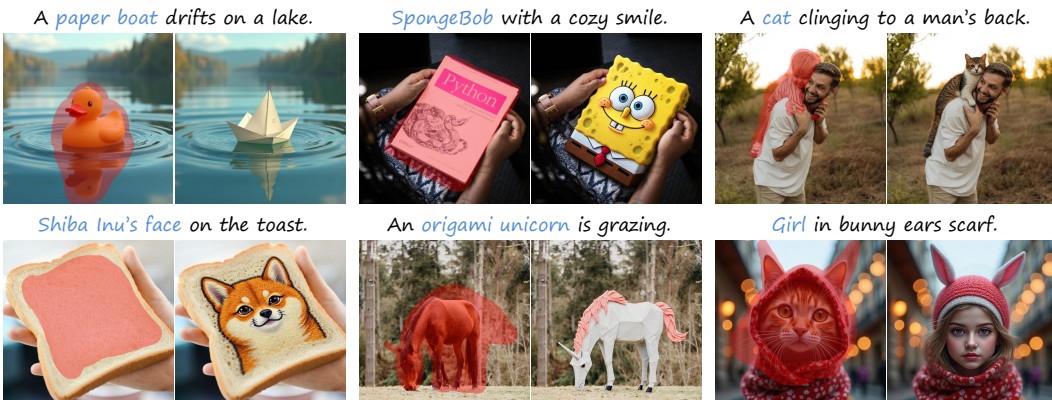

Figure 1: **Results of our model.** The inpainting outputs generated by our model are visually coherent, semantically aligned with text prompts, and consistent with human aesthetic preferences.

## ABSTRACT

This paper investigates image inpainting with preference alignment. Instead of introducing a novel method, we go back to basics and revisit fundamental problems in achieving such alignment. We leverage the prominent direct preference optimization approach for alignment training and employ public reward models to construct preference training datasets. Experiments are conducted across nine reward models, two benchmarks, and two baseline models with varying structures and generative algorithms. Our key findings are as follows: (1) Most reward models deliver valid reward scores for constructing preference data, even if some of them are not reliable evaluators. (2) Preference data demonstrates robust trends in both candidate scaling and sample scaling across models and benchmarks. (3) Observable biases in reward models, particularly in brightness, composition, and color scheme, render them susceptible to cause reward hacking. (4) A simple ensemble of these models yields robust and generalizable results by mitigating such biases. Built upon these observations, our alignment models significantly outperform prior models across standard metrics, GPT-4 assessments, and human evaluations, without any changes to model structures or the use of new datasets. We hope our work can set a simple yet solid baseline, pushing this promising frontier. Our code is available at: https://github.com/shenytzzz/Follow-Your-Preference.

## 1 INTRODUCTION

Image inpainting (Bertalmio et al., 2000) aims to fill in user-specified regions of an image in a visually coherent and realistic manner. It holds great value in applications such as photo restoration (Liang et al., 2021), content creation (Zhuang et al., 2024), and image editing (Zhang et al., 2023a). With the unprecedented success of diffusion models (Ho et al., 2020) and flow-based models (Lipman et al., 2022), image inpainting has become a prominent research focus in recent years.

---

[*] Equal contribution.
[†] Corresponding author.

Aligning human preferences in visual generation has emerged as a focal point of research efforts (Wallace et al., 2024; Xue et al., 2025; Black et al., 2023; Fan et al., 2023). While great progress has been made in image inpainting (Wu et al., 2025; Ju et al., 2024; Zhuang et al., 2024; Manukyan et al., 2023), research on aligning inpainting results with human preferences remains limited.

This paper explores image inpainting with preference alignment. Given the limited work on this task, our goal is not to present a novel method, but rather to rethink foundational questions. To this end, we adopt the prominent Direct Preference Optimization (DPO) (Rafailov et al., 2023; Wallace et al., 2024; Liu et al., 2025) to conduct studies due to its simplicity and efficiency. Instead of relying on costly and non-scalable human annotations, we employ public, off-the-shelf reward models for constructing preference training datasets. Our study focuses on several key questions: (1) How *effective* are these reward models in scoring and constructing high-quality preference data? (2) How *scalable* is preference data with respect to the candidate quantity and the sample quantity? (3) How does *reward hacking* (Pan et al., 2022) occur, and what method can be used to mitigate it?

To answer these questions, we conduct experiments across *nine widely used reward models* (e.g., HPSv2 (Wu et al., 2023), PickScore (Kirstain et al., 2023)), *two representative evaluation benchmarks* (BrushBench (Ju et al., 2024), EditBench (Wang et al., 2023)), and *two baseline inpainting models* (BrushNet (Ju et al., 2024), FLUX.1 Fill (BlackForestLabs, 2024b)) with diverse architectures (U-Net (Ronneberger et al., 2015), Transformer-based (Vaswani et al., 2017)) and generative algorithms (diffusion (Ho et al., 2020), flow-based (Lipman et al., 2022)). Our findings reveal that: **(1)** Most reward models provide *valid* reward signals for constructing effective preference training data, despite some being unreliable as evaluators and exhibiting shared biases. **(2)** Preference data shows *consistent trends* in both candidate scaling and sample scaling across baseline models and benchmarks. However, biases in certain reward models (e.g., HPSv2) can lead to reward hacking, undermining scaling effectiveness. **(3)** We identify explicit biases in reward models—particularly in *brightness*, *composition*, and *color scheme*—making them vulnerable to reward hacking. For example, HPSv2 tends to favor images with bright lighting, complex composition with rich details, and vivid colors; PickScore shows the opposite tendency. We also find that BrushNet generates vibrant images, making PickScore suitable for it; while FLUX.1 Fill produces plain images, aligning well with the property of HPSv2. **(4)** A simple ensemble of these reward models exhibits strong versatility across models, producing balanced and aesthetically pleasing inpainting results.

Building on these observations, we propose simple yet effective models via reward ensemble. Without modifying model architectures or introducing new datasets, our models substantially outperform state-of-the-art models—across standard metrics, GPT-4 assessments, and human evaluations. Visualizations show that our models generate more coherent and visually appealing results than competitors. We hope our work can establish a simple yet strong baseline to advance this research field.

## 2 RELATED WORK

**Image inpainting** (Bertalmio et al., 2000) is the process of filling in missing regions of an image. Great progress (Manukyan et al., 2023; Wang et al., 2025c; Yuan et al., 2023c;b) has been achieved by employing diffusion (Ho et al., 2020) and flow-based (Lipman et al., 2022; Esser et al., 2024; Chen et al., 2024; Ma et al., 2024b) models, which iteratively transform random noise into structured data. Some pioneering efforts make attempts (Avrahami et al., 2023; Rombach et al., 2022; Zhang et al., 2023b) to incorporate these generative models into image inpainting, and others (Ju et al., 2024; Zhuang et al., 2024; Yuan et al., 2023a) later advance and refine it. For example, BrushNet (Ju et al., 2024) is a dual-branch model that decouples masked image feature extraction from generation. FLUX (BlackForestLabs, 2024a) is a rectified flow transformer that generates impressive images. Its variant, FLUX.1 Fill (BlackForestLabs, 2024b), is tailored specifically for image inpainting.

**Image generation with preference alignment** is an emerging field that seeks to align synthesized images with human preferences (Hu et al., 2026; Yuan et al., 2022). Some previous works (Black et al., 2023; Fan et al., 2023) employ reinforcement learning (RL) (Sutton et al., 1998) to fine-tune the models using feedback derived from reward models. Recent works have begun exploring methods based on Direct Preference Optimization (DPO) (Rafailov et al., 2023), without the RL process. For instance, Diffusion-DPO (Wallace et al., 2024) is the first attempt to align the generation with human preferences by optimizing diffusion-based pairwise feedback. PrefPaint (Liu et al., 2024b) refines image inpainting results by using a reward model trained on human-annotated data.

## 3 PRELIMINARIES

### 3.1 DIFFUSION MODELS AND FLOW-BASED MODELS

Diffusion models (Sohl-Dickstein et al., 2015; Song et al., 2020), such as **DDPM** (Ho et al., 2020), are a class of generative models that learn to reverse a gradual noise corruption process. DDPM assumes a forward process that gradually applies noise to real data. At timestep $t$, the real data $x_0$ is destroyed to $x_t$: $q(x_t|x_0) = \mathcal{N}(x_t; \sqrt{\bar{\alpha}_t}x_0, (1 - \bar{\alpha}_t)\mathbf{I})$, where $\bar{\alpha}_t$ is noise scheduling hyper-parameters. It has a reparameterization formula: $x_t = \sqrt{\bar{\alpha}_t}x_0 + \sqrt{1 - \bar{\alpha}_t}\epsilon$, where noise $\epsilon \sim \mathcal{N}(0, \mathbf{I})$. DDPM learns a reverse process using a denoising model $\epsilon_\theta$ with parameters $\theta$, inverting the forward process: $p_\theta(x_{t-1}|x_t) = \mathcal{N}(\mu_\theta(x_t), \Sigma_\theta(x_t))$. The denoising model $\epsilon_\theta$ can be trained by minimizing:

$$\mathcal{L}_{\text{DDPM}} = \mathbb{E}_{t,x_0,\epsilon}\left[||\epsilon - \epsilon_\theta(\sqrt{\bar{\alpha}_t}x_0 + \sqrt{1 - \bar{\alpha}_t}\epsilon, t)||^2\right]. \tag{1}$$

Flow-based models (Esser et al., 2024) are generative models that learn to model data distributions using invertible transformations. Recently, **Flow Matching** (Lipman et al., 2022) has emerged as a prominent approach for visual generation (Ma et al., 2024a; Gao et al., 2025a). It usually learns a continuous-time flow that transforms a simple prior distribution into the data distribution by solving an ODE. The process, with an optimal-transport path, employs a linear interpolation scheme: $x_t = (1 - t)x_0 + t\epsilon$. A denoising model $v_\theta$ is trained to predict the velocity field by minimizing:

$$\mathcal{L}_{\text{FlowMatching}} = \mathbb{E}_{t,x_0,\epsilon}\left[||v_\theta((1 - t)x_0 + t\epsilon, t) - (\epsilon - x_0)||^2\right]. \tag{2}$$

**U-Net** (Ronneberger et al., 2015) is used as the basic model structure by many previous denoising models (Ho et al., 2020; Song et al., 2020). U-Net is a symmetric encoder-decoder architecture that captures multi-scale features through progressive downsampling and upsampling. **Transformers** (Vaswani et al., 2017), employed in recent works (Gao et al., 2025b;a; Wan et al., 2025; Kong et al., 2024), process all data elements in parallel using attention, facilitating training scalibility.

To improve the reliability and generalization of conclusions drawn in our studies, we conduct investigations using two different **baseline models—BrushNet** (Ju et al., 2024) and **FLUX.1 Fill** (Black-ForestLabs, 2024b), introduced in section 2. BrushNet is built on a U-Net-like architecture and trained with the DDPM loss, while FLUX leverages transformers and learns via Flow Matching.

### 3.2 PREFERENCE ALIGNMENT

The standard pipeline for training large-scale models typically involves pre-training, supervised fine-tuning, and preference alignment. Preference alignment refines model outputs to better match human values. Reinforcement Learning from Human Feedback (RLHF) (Bai et al., 2022) is a popular alignment approach. It utilizes human preferences on model outputs to train a *separate reward model*, which subsequently provides rewards for alignment via reinforcement learning algorithms such as PPO (Schulman et al., 2017) and GRPO (Guo et al., 2025). In comparison, **Direct Preference Optimization (DPO)** (Rafailov et al., 2023), which performs direct supervised learning, offers higher training efficiency. It constructs a preference dataset that comprises *preferred samples* and *dispreferred samples*. DPO learns human preferences implicitly contained within the data by maximizing:

$$\mathbb{E}_{x,y^w,y^l}[\log \sigma(\beta \log \frac{\pi_\theta(y^w|x)}{\pi_{\text{ref}}(y^w|x)} - \beta \log \frac{\pi_\theta(y^l|x)}{\pi_{\text{ref}}(y^l|x)})], \tag{3}$$

where $\sigma$ is the sigmoid function; $\pi_\theta$ and $\pi_{\text{ref}}$ are the *policy* and the *reference policy* respectively. In image generation, given a text prompt $x$, $y^w$ and $y^l$ denote the generated preferred image and dispreferred image, respectively. The hyper-parameter $\beta$ controls the strength of regularization: a large value of $\beta$ increases regularization pressure, dampening preference learning. In visual generation, Equation 3 can be derived to yield a simplified loss (Wallace et al., 2024; Liu et al., 2025):

$$\mathcal{L}_{\text{DPO}} = -\mathbb{E}[\log \sigma(-\beta((\mathcal{L}_\theta^w - \mathcal{L}_{\text{ref}}^w) - (\mathcal{L}_\theta^l - \mathcal{L}_{\text{ref}}^l)))], \tag{4}$$

where $\mathcal{L}_\theta^w$ and $\mathcal{L}_\theta^l$ denote the loss (Equation 1 or Equation 2) applied to the policy on preferred samples and dispreferred samples, respectively; similarly, $\mathcal{L}_{\text{ref}}^w$ and $\mathcal{L}_{\text{ref}}^l$ denote the loss applied to the reference policy. This loss function aligns the distribution of generated samples with the preferred data distribution and diverges from the dispreferred distribution. Due to the simplicity, efficiency, and stability of DPO, *this paper will explore preference alignment for image inpainting by optimizing Equation 4 on different preference datasets that are constructed for investigation.*

Table 1: Comparisons of reward models using **BrushNet** on BrushBench and EditBench.

| reward model | CLIPScore | | Aesthetic | | ImageR | | PickScore | | HPSv2 | | VQAScore | | UnifiedR | | Perception | | HPSv3 | | GPT-4 | |
|---|---|---|---|---|---|---|---|---|---|---|---|---|---|---|---|---|---|---|---|---|
| | Brush. | Edit. | Brush. | Edit. | Brush. | Edit. | Brush. | Edit. | Brush. | Edit. | Brush. | Edit. | Brush. | Edit. | Brush. | Edit. | Brush. | Edit. | Brush. | Edit. |
| Baseline | 26.415 | 27.337 | 6.425 | 5.392 | 12.717 | -1.296 | 22.133 | 20.616 | 27.509 | 23.076 | 9.060 | 6.770 | 3.303 | 2.100 | 26.290 | 26.410 | 5.749 | 0.403 | 79.391 | 57.046 |
| Random | 26.441 | 27.631 | 6.424 | 5.392 | 12.685 | -1.136 | 22.130 | 20.642 | 27.501 | 23.067 | 9.050 | **6.917** | 3.302 | 2.110 | 26.292 | 26.422 | 5.738 | 0.425 | 79.177 | 56.753 |
| CLIPScore | 26.461 | **27.710** | 6.430 | 5.393 | 12.782 | -0.720 | 22.146 | 20.680 | 27.582 | 23.316 | 9.062 | 6.894 | 3.341 | 2.124 | 26.324 | 26.548 | 5.777 | 0.640 | 79.661 | 57.539 |
| Aesthetic | 26.465 | 27.355 | 6.477 | **5.520** | 12.994 | -0.877 | 22.221 | 20.689 | 27.594 | 23.166 | 9.065 | 6.828 | 3.343 | 2.140 | 26.293 | 26.342 | 5.922 | 0.597 | 81.603 | 58.603 |
| ImageR | 26.471 | 27.539 | 6.462 | 5.434 | 12.891 | -0.377 | 22.153 | 20.701 | 27.672 | 23.467 | 9.036 | 6.761 | 3.334 | 2.144 | 26.305 | 26.501 | 5.913 | 0.782 | 80.341 | 57.806 |
| PickScore | 26.397 | 27.199 | 6.454 | 5.454 | 12.893 | -1.364 | **22.254** | 20.732 | 27.322 | 22.933 | 9.062 | 6.873 | 3.353 | **2.178** | 26.273 | 26.427 | 5.750 | 0.469 | **82.726** | **59.550** |
| HPSv2 | 26.481 | 27.677 | 6.476 | 5.495 | 12.890 | **0.128** | 22.137 | 20.725 | **27.818** | 23.742 | 9.073 | 6.818 | 3.332 | 2.155 | 26.361 | 26.678 | 5.979 | **1.061** | 79.914 | 57.658 |
| VQAScore | 26.442 | 27.524 | 6.429 | 5.407 | 12.658 | -0.800 | 22.126 | 20.667 | 27.527 | 23.234 | 9.038 | 6.879 | 3.311 | 2.139 | 26.326 | 26.406 | 5.723 | 0.555 | 78.877 | 56.975 |
| UnifiedR | 26.428 | 27.505 | 6.433 | 5.402 | 12.764 | -0.812 | 22.157 | 20.675 | 27.562 | 23.204 | 9.061 | 6.857 | 3.329 | 2.155 | 26.320 | 26.436 | 5.800 | 0.540 | 80.333 | 57.185 |
| Perception | 26.448 | 27.484 | 6.428 | 5.393 | 12.789 | -0.973 | 22.177 | 20.660 | 27.519 | 23.111 | 9.069 | 6.894 | 3.327 | 2.160 | 26.310 | 26.515 | 5.764 | 0.433 | 80.277 | 57.254 |
| HPSv3 | 26.461 | 27.547 | 6.464 | 5.448 | 12.922 | -0.146 | 22.176 | 20.713 | 27.758 | 23.491 | 9.065 | 6.850 | 3.344 | 2.158 | 26.317 | 26.535 | 6.014 | 0.863 | 80.623 | 57.485 |
| Ensemble | **26.535** | 27.398 | **6.485** | 5.497 | **13.037** | -0.352 | 22.229 | **20.735** | 27.797 | 23.522 | 9.053 | 6.892 | **3.365** | 2.176 | 26.338 | 26.603 | **6.074** | 1.015 | 82.172 | 58.986 |

**Bold** values denote the best results. Underlined values denote the second-best results. Values in blue denote the results below the baseline or random chance.

## 3.3 Reward Models

Reward models play an important role in preference alignment: they provide real-time rewards in RLHF (Schulman et al., 2017), and offer scores for constructing offline preference data in DPO (Wang et al., 2025b; Lee et al., 2025). However, prior works (Wang et al., 2025a; Xue et al., 2025) directly employ off-the-shelf reward models for visual preference alignment without sufficient evaluations. In this paper, we evaluate the effectiveness of these reward models in constructing preference data via extensive studies. Specifically, we examine the following **public reward models**: (1) **CLIPScore** (Hessel et al., 2021) measures semantic alignment between images and text prompts by calculating cosine similarities of their CLIP embeddings (Radford et al., 2021). (2) **Aesthetic** (Schuhmann et al., 2022) predicts human aesthetic preferences on top of the CLIP embeddings. (3) **ImageReward (ImageR)** (Xu et al., 2023) is trained by fine-tuning BLIP (Li et al., 2022) on 137K preference samples. (4) **PickScore** (Kirstain et al., 2023) is a CLIP-based image scoring model, trained on over 500K synthesized image samples with users' preference choices. (5) **HPSv2** (Wu et al., 2023) is also a CLIP-based model that evaluates both image quality and text-image alignment by learning from 798K human preferences on 433K sample pairs. (6) **VQAScore** (Lin et al., 2024) provides a semantic alignment score by computing the probability of a VQA model answering "yes" to each question: "Does this figure show {text}?". (7) **UnifiedReward (UnifiedR)** (Wang et al., 2025b) is a unified model that assesses both visual generation and understanding. (8) **Perception Encoder (Perception)** (Bolya et al., 2025) is trained by contrastive visual-language pre-training, producing semantically aligned multimodal embeddings. (9) **HPSv3** (Ma et al., 2025b) is trained on 1.5M annotated sample pairs using Qwen2VL-7B (Wang et al., 2024).

To assess their efficacy, these models are employed to assign reward scores to candidate samples which are generated by the baseline models with different random seeds. The resulting highest- and lowest-scoring samples from each text prompt are subsequently utilized as the preferred and dispreferred samples for DPO training. Based on the evaluation of training results, the top-performing ones are designated as the most effective reward models to provide accurate rewards, and vice versa.

## 4 How Effective are Reward Models?

The ability of reward models to accurately predict human preferences is critical to the performance of preference alignment algorithms. To evaluate this capability, we apply DPO on the preference data constructed by the reward models and evaluate the model'performance after training. Specifically, based on the popular dataset of BrushData (Ju et al., 2024), we generate 16 **candidate** inpainting results with varied random seeds for each prompt and the corresponding masked image. The candidates are scored by the reward models, and the highest-scoring (preferred) and lowest-scoring (dispreferred) samples form preference pairs for DPO training. Following (Ma et al., 2025a; Wang et al., 2025b), the reward models are employed to serve two purposes: (1) *providing scores to construct training data*, and (2) *evaluating performance after training*. All experiments adhere to the same training configurations by default (e.g., learning rate is 1e-7, $\beta$ is 2000, 2000 training steps, etc.), with the only variation being the reward model used to score and construct the training data.

Table 2: Comparisons of reward models using **FLUX.1 Fill** on BrushBench and EditBench.

| reward model | CLIPScore | | Aesthetic | | ImageR | | PickScore | | HPSv2 | | VQAScore | | UnifiedR | | Perception | | HPSv3 | | GPT-4 | |
|---|---|---|---|---|---|---|---|---|---|---|---|---|---|---|---|---|---|---|---|---|
| | Brush. | Edit. | Brush. | Edit. | Brush. | Edit. | Brush. | Edit. | Brush. | Edit. | Brush. | Edit. | Brush. | Edit. | Brush. | Edit. | Brush. | Edit. | Brush. | Edit. |
| Baseline | 26.244 | 27.103 | 6.429 | 5.458 | 12.760 | 4.910 | 22.327 | 21.211 | 27.476 | 24.076 | 9.081 | 8.012 | 3.360 | 2.485 | 25.945 | 26.834 | 6.055 | 2.470 | 83.935 | 66.979 |
| Random | 26.239 | 27.078 | 6.431 | 5.459 | 12.772 | 4.955 | 22.328 | 21.211 | 27.475 | 24.100 | 9.077 | 8.030 | 3.356 | 2.491 | 25.944 | 26.838 | 6.056 | 2.490 | 83.517 | 66.942 |
| CLIPScore | 26.233 | 27.072 | 6.432 | 5.477 | 12.791 | 4.997 | 22.329 | 21.215 | 27.487 | 24.121 | 9.071 | **8.071** | 3.361 | 2.512 | 25.948 | 26.859 | 6.056 | 2.499 | 83.942 | 66.997 |
| Aesthetic | 26.250 | 27.200 | 6.432 | 5.478 | 12.823 | 5.175 | 22.337 | 21.219 | 27.520 | 24.142 | 9.075 | 8.001 | 3.363 | 2.507 | 25.954 | 26.878 | 6.075 | 2.577 | 83.950 | 67.906 |
| ImageR | **26.251** | 27.121 | 6.434 | **5.481** | 12.823 | 5.001 | 22.336 | 21.211 | 27.518 | 24.143 | 9.080 | 7.977 | 3.362 | 2.536 | 25.946 | 26.846 | 6.078 | 2.550 | 84.176 | 67.785 |
| PickScore | 26.236 | 27.195 | 6.436 | 5.476 | 12.879 | 5.134 | 22.341 | 21.223 | 27.530 | 24.154 | 9.076 | 8.003 | **3.383** | 2.514 | 25.955 | 26.900 | 6.105 | 2.548 | 84.188 | 67.100 |
| HPSv2 | 26.246 | 27.160 | 6.441 | 5.475 | **12.904** | 5.145 | **22.356** | **21.232** | **27.605** | **24.202** | **9.085** | 8.028 | 3.363 | **2.553** | 25.963 | 26.895 | **6.181** | **2.605** | **84.699** | **68.186** |
| VQAScore | 26.243 | 27.101 | 6.432 | 5.466 | 12.781 | 4.926 | 22.329 | 21.215 | 27.486 | 24.104 | 9.075 | 8.020 | 3.353 | 2.501 | 25.950 | 26.861 | 6.046 | 2.473 | 83.854 | 66.793 |
| UnifiedR | 26.235 | 27.133 | 6.434 | 5.473 | 12.769 | 5.034 | 22.331 | 21.212 | 27.485 | 24.120 | 9.076 | 8.014 | 3.366 | 2.517 | 25.954 | 26.853 | 6.057 | 2.524 | 83.950 | 67.372 |
| Perception | **26.251** | 27.088 | 6.433 | 5.466 | 12.752 | 4.975 | 22.331 | 21.209 | 27.479 | 24.109 | 9.082 | 8.023 | 3.356 | 2.514 | 25.951 | 26.818 | 6.064 | 2.504 | 84.022 | 68.024 |
| HPSv3 | 26.238 | **27.223** | 6.441 | 5.465 | 12.855 | 5.092 | 22.340 | 21.226 | 27.534 | 24.155 | 9.083 | 8.000 | 3.378 | 2.497 | 25.957 | 26.859 | 6.106 | 2.568 | 84.615 | 68.107 |
| Ensemble | 26.239 | 27.158 | **6.442** | 5.472 | 12.884 | **5.239** | 22.346 | 21.215 | 27.560 | 24.146 | 9.082 | 8.036 | 3.367 | 2.507 | **25.963** | 26.905 | 6.151 | 2.577 | 84.628 | 67.549 |

**Bold** values denote the best results. Underlined values denote the second-best results. Values in blue denote the results below the baseline or random chance.

Note that we may encounter an **oracle reward model** (Ma et al., 2025a), where the same model is used both for data construction and performance evaluation within one experiment. We assess results on two benchmarks, i.e., BrushBench (Ju et al., 2024) and EditBench (Wang et al., 2023).

Here, we introduce a new reward model—**Ensemble** that constructs samples based on the average ranking of all the reward models. We make **GPT-4** (Achiam et al., 2023) serve as a "fair" evaluator by assessing aesthetic quality, structural coherence, and semantic alignment of the results (see details in Appendix). We report two other results. **Baseline:** The model's performance prior to DPO training. **Random:** It involves training with randomly sampled preferred and dispreferred pairs. Results are reported in Table 1 and Table 2. We have the following observations and conclusions.

**Some reward models are not reliable evaluators.** It is believed that an accurate and robust reward model should assign high evaluation scores to models trained on its own preference dataset (i.e., the oracle reward model setting). Surprisingly, we find that CLIPScore, VQAScore, and Perception fail to meet this requirement—in Table 2, their scores can be even lower than baseline or random results. We hypothesize that the fail of CLIPScore and Perception stems from their large-scale yet potentially coarse contrastive pre-training; the fail of VQAScore likely arises from its simplistic, VQA-like evaluation approach. In light of this, *we exclude these models from subsequent analyses*.

**Most reward models provide valid reward scores.** Most reward models are capable of offering valid reward scores for preference data construction, as they outperform both the baseline and random selection across most evaluation results—especially GPT-4. Even though CLIPScore and Perception are observed to be less effective at accurately evaluating on small-scale benchmarks, they remain viable when their reward scores are incorporated into larger-scale preference training datasets. In this context, we continue to attribute VQAScore's limitations to its simple scoring methodology.

**Reward models may share common biases.** We find that the model trained on HPSv2-constructed data outperforms most competitors when evaluated using public reward models. Specifically, when trained using BrushNet, it ranks first or second in 4 out of 12 evaluations; when trained using FLUX.1 Fill, it ranks first or second in 9 out of 12 evaluations. This pattern aligns with GPT-4's results when using FLUX.1 Fill but diverges when using BrushNet—under the latter condition, the model is largely outperformed by PickScore. We posit that HPSv2 and many other models may share some common biases, which can potentially lead to reward hacking (Pan et al., 2022).

**Ensemble is an accurate and robust reward model.** It shows that Ensemble ranks first or second in 11 out of 12 public model evaluations when using BrushNet, and 7 out of 12 when using FLUX.1 Fill. Besides, Ensemble ranks first or second in 3 out of 4 GPT-4's evaluations across both baseline models, demonstrating its robustness in constructing effective preference data. We hypothesize that its versatility arises from the bias of reward models being weakened in Ensemble.

**Discussion.** Part of the above analysis is based on an untested assumption—GPT-4 is an ideal evaluator. We will examine its validity as well as reward hacking in section 6 and the Appendix.

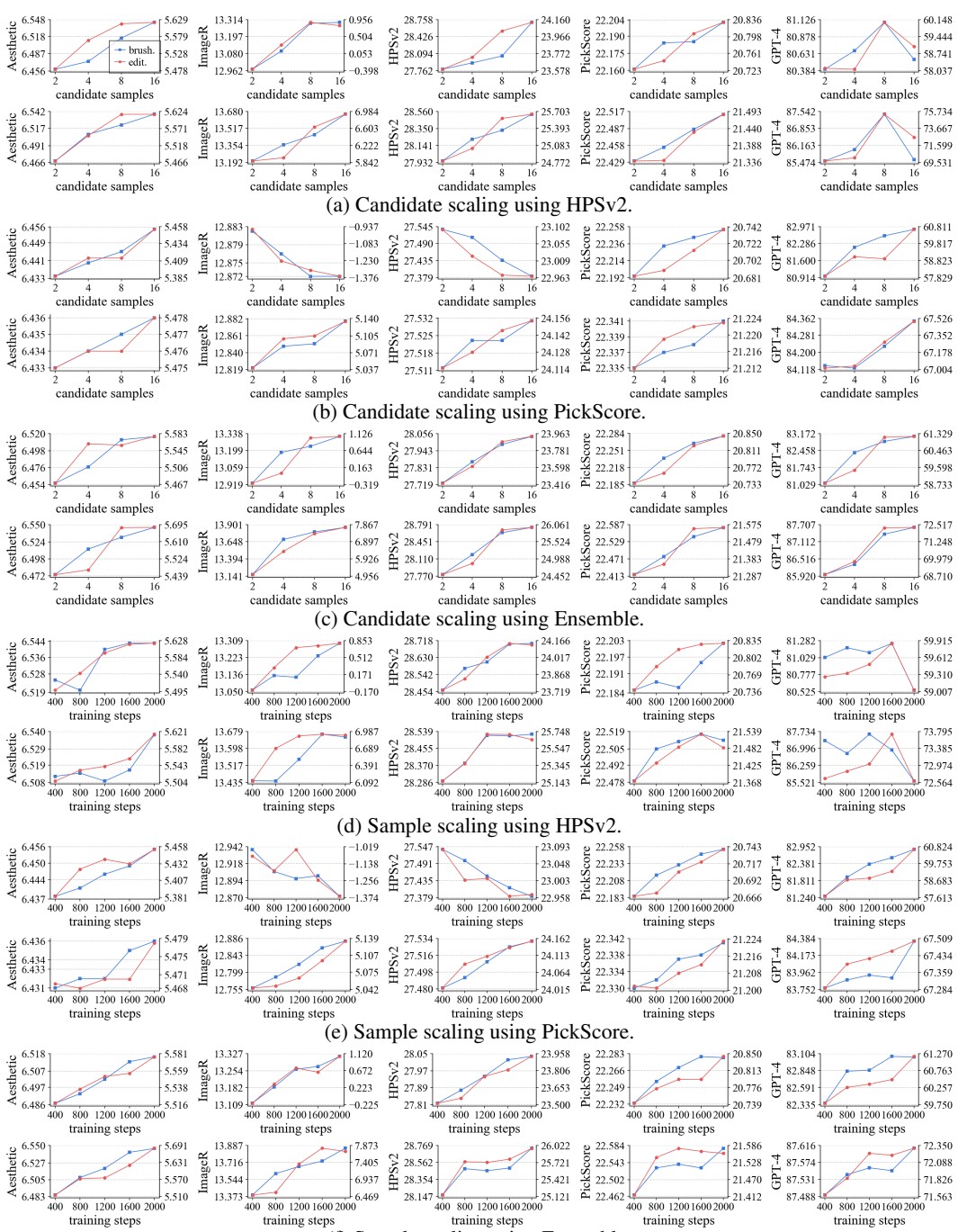

Figure 2: **Candidate scaling** (a-c) and **sample scaling** (d-f) using HPSv2, PickScore, and Ensemble. We employ Aesthetic, ImageR, HPSv2, PickScore, and GPT-4 for evaluation. The first and second row of each sub-figure is based on BrushNet and FLUX.1 Fill, respectively. We use training steps to indicate the consumed samples to align the scaling across models (their batch-sizes are different).

## 5 HOW SCALABLE ARE PREFERENCE DATA?

The results in section 4 have shown that *HPSv2*, *PickScore*, and *Ensemble* are promising reward models. Building on this finding, we conduct an investigation into the scalability of preference data using these reward models. Specifically, we explore along two dimensions: (1) **Candidate scaling.** As the number of candidate samples generated from different random seeds increases, their diversity expands. This augmentation in diversity would enhance the accuracy of the construction of preferred

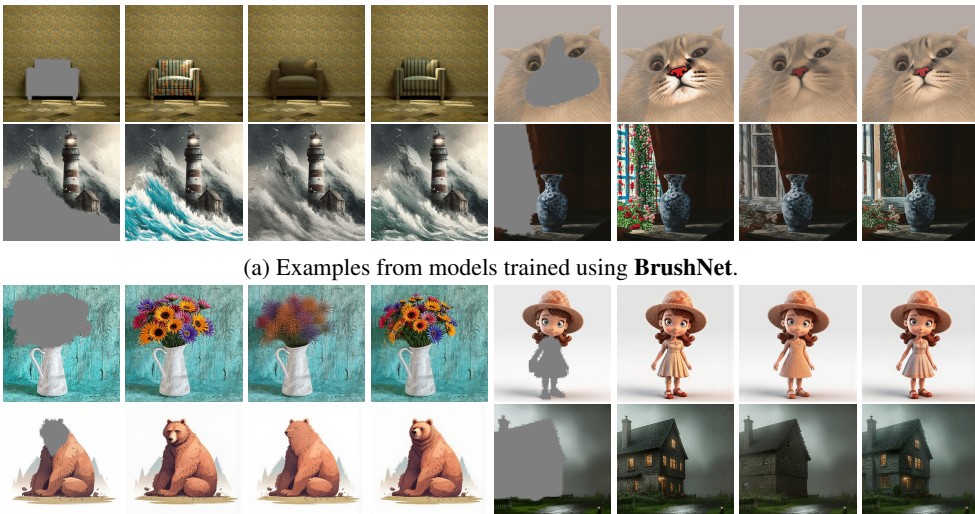

(a) Examples from models trained using **BrushNet**.

(b) Examples from models trained using **FLUX.1 Fill**.

Figure 3: **Bias studies.** In each sub-figure, the four images (from left to right) display: the *masked image*, followed by inpainting results from models trained using *HPSv2*, *PickScore*, and *Ensemble*. We omit text prompts for brevity. Zoom in to see details. Find more examples in the Appendix.

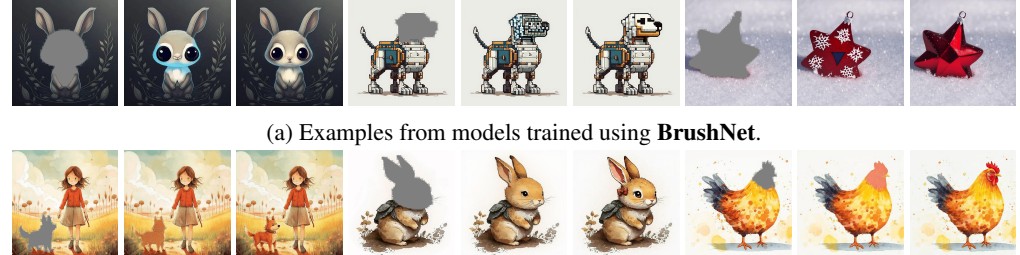

(a) Examples from models trained using **BrushNet**.

(b) Examples from models trained using **FLUX.1 Fill**.

Figure 4: **Qualitative results of ablations.** In each sub-figure, the three images (from left to right) display: the *masked image*, followed by inpainting results from *baseline models* and *baseline models + preference alignment using Ensemble*. We omit text prompts for brevity. Zoom in to seed details.

and dispreferred samples by enhancing their differences. (2) **Sample scaling.** A larger dataset enables the model to capture nuanced patterns more comprehensively, leading to deeper learning of preferences. Based on the insights in section 4, we select Aesthetic, ImageR, HPSv2, PickScore, and GPT-4 as the evaluation models. To enable the model to achieve optimal performance, we conduct a search over two typical hyper-parameters—$\beta$ and learning rate (see details in the Appendix), before the scaling experiments. For each experiment, we tune one scaling dimension and fix the other dimension. The results are reported in Figure 2. We have the following results and discoveries.

**Consistent scaling trends across models and benchmarks.** First, we observe that data scaling demonstrates robust trends regardless of the model used—BrushNet and FLUX.1 Fill, in the first and second rows of each sub-figure, respectively. Second, we find that similar scaling trends emerge when evaluating on different benchmarks, as evidenced by the comparable patterns of the two lines within each sub-figure. These findings indicate that the observed scaling behavior is robust and generalizable. However, we also observe some inconsistent phenomena: when using PickScore as the reward model, ImageR/HPSv2 exhibit opposite trends on BrushNet and FLUX.1 Fill. This issue is caused by the characteristics of both reward models and baseline models, as analyzed in section 6.

**Reward hacking from HPSv2 undermines training.** When evaluated by Aesthetic, ImageR, HPSv2, and PickScore, using HPSv2 as the reward model shows benefits from both candidate scaling and sample scaling. However, its GPT-4 results deteriorate significantly in the later stages of scaling. This observation aligns with our finding in section 4, where HPSv2 achieves good results under public model evaluations but sometimes loses to others when assessed by GPT-4. We hypothesize that this degradation stems from some shared common biases among these reward models.

Table 3: Ablation studies on a new dataset of *I Dream My Painting* (Fanelli et al., 2025).

| inpainting model | CLIPScore | Aesthetic | ImageR | PickScore | HPSv2 | VQAScore | UnifiedR | Perception | HPSv3 | GPT-4 |
|---|---|---|---|---|---|---|---|---|---|---|
| BrushNet | 24.849 | 5.923 | -0.246 | 20.550 | 19.749 | 8.503 | 2.317 | 27.317 | -0.551 | 72.669 |
| **BruPA (ours)** | **25.460** | **6.111** | **2.152** | **20.735** | **21.086** | **8.653** | **2.463** | **28.294** | **1.265** | **73.739** |
| FLUX.1 Fill | 24.194 | 6.017 | 0.544 | 20.855 | 20.203 | 8.667 | 2.476 | 26.627 | 0.547 | 76.391 |
| **FluPA (ours)** | **25.500** | **6.448** | **5.961** | **21.407** | **23.770** | **9.031** | **2.784** | **28.868** | **5.023** | **79.255** |

**Bold** values denote the best results.

Table 4: Comparisons of state-of-the-art image inpainting models on BrushBench and EditBench.

| inpainting model | CLIPScore | | Aesthetic | | ImageR | | PickScore | | HPSv2 | | VQAScore | | UnifiedR | | Perception | | HPSv3 | | GPT-4 | |
|---|---|---|---|---|---|---|---|---|---|---|---|---|---|---|---|---|---|---|---|---|
| | Brush. | Edit. | Brush. | Edit. | Brush. | Edit. | Brush. | Edit. | Brush. | Edit. | Brush. | Edit. | Brush. | Edit. | Brush. | Edit. | Brush. | Edit. | Brush. | Edit. |
| SDI | 26.304 | 26.526 | 6.368 | 5.377 | 12.026 | -1.100 | 22.105 | 20.791 | 27.079 | 23.203 | 8.981 | 6.923 | 3.268 | 2.069 | 26.190 | 25.382 | 5.320 | 0.849 | 79.004 | 60.751 |
| CNI | 26.341 | 26.972 | 6.305 | 5.382 | 11.421 | -1.044 | 21.953 | 20.874 | 26.633 | 23.076 | 8.890 | 6.906 | 3.218 | 2.125 | 26.150 | 25.894 | 4.546 | 0.894 | 74.173 | 63.921 |
| BLD | 26.337 | 27.666 | 6.262 | 5.372 | 11.161 | 0.563 | 21.901 | 20.980 | 26.723 | 23.839 | 8.852 | 7.467 | 3.202 | 2.228 | 26.128 | 27.093 | 4.559 | 1.114 | 71.794 | 62.690 |
| PowerPoint | 26.265 | 27.291 | 6.312 | 5.448 | 11.771 | 0.720 | 22.089 | 20.912 | 27.065 | 23.347 | 8.931 | 7.238 | 3.271 | 2.219 | 26.123 | 26.264 | 5.112 | 1.068 | 78.241 | 63.092 |
| PrefPaint | 26.268 | 25.569 | 6.377 | 5.296 | 11.798 | -3.023 | 22.125 | 20.666 | 26.855 | 22.241 | 8.925 | 6.226 | 3.271 | 1.951 | 26.116 | 24.264 | 5.208 | -0.336 | 80.327 | 60.815 |
| StrDiffusion | 23.872 | 21.398 | 5.330 | 4.405 | -0.063 | -16.282 | 20.342 | 19.147 | 21.431 | 16.616 | 7.281 | 3.662 | 2.417 | 1.236 | 23.381 | 20.102 | -2.243 | -7.131 | 34.255 | 25.200 |
| HD-Painter | 26.367 | 26.934 | 6.480 | 5.640 | 12.913 | 0.046 | 22.314 | 21.016 | 27.931 | 23.951 | 9.019 | 6.682 | 3.349 | 2.136 | 26.214 | 25.721 | 6.224 | 1.983 | 85.016 | 69.087 |
| ASUKA | 24.387 | 20.842 | 6.294 | 5.078 | 5.208 | -14.110 | 21.601 | 19.603 | 25.285 | 18.702 | 7.681 | 3.631 | 2.862 | 1.282 | 23.959 | 19.017 | 3.556 | -3.506 | 75.140 | 58.686 |
| BrushNet | 26.415 | 27.337 | 6.425 | 5.392 | 12.717 | -1.296 | 22.133 | 20.616 | 27.509 | 23.076 | 9.060 | 6.770 | 3.303 | 2.100 | 26.290 | 26.410 | 5.749 | 0.403 | 79.391 | 57.046 |
| **BruPA (ours)** | **26.547** | 27.694 | 6.516 | 5.577 | 13.315 | **10.463** | 22.279 | 20.844 | 28.037 | 23.933 | 9.093 | 7.043 | 3.371 | 2.193 | **26.390** | 26.881 | 6.276 | 1.398 | 83.054 | 61.186 |
| FLUX.1 Fill | 26.244 | 27.103 | 6.429 | 5.458 | 12.760 | 4.910 | 22.327 | 21.211 | 27.476 | 24.076 | 9.081 | 8.021 | 3.360 | 2.485 | 25.945 | 26.834 | 6.055 | 2.470 | 83.935 | 66.979 |
| **FluPA (ours)** | 26.436 | 27.813 | 6.546 | 5.681 | 13.859 | 7.707 | 22.577 | 21.559 | 28.735 | 25.972 | 9.152 | 8.434 | 3.457 | 2.649 | 26.096 | 27.617 | 7.000 | 4.230 | 87.609 | 72.307 |

**Bold** values denote the best results. Underlined values denote the second-best results. All methods are evaluated using official implementations with blending (Ju et al., 2024).

**Ensemble offers robust data scaling by resisting hacking.** Although PickScore demonstrates good scaling behavior, its performance remains sub-optimal. In contrast, the Ensemble approach achieves the best results across benchmarks, model structures, evaluation models, and scaling dimensions. This is likely because Ensemble averages the preference choices of different reward models, which eliminates the biases of the employed reward models and improves its resistance to the hacking.

## 6    HOW REWARD HACKING HAPPENS?

We identify potential biases in reward models that may lead to reward hacking, as discussed in section 4 and section 5. In this section, we delve deeper into exploring these intriguing biases— examining their nature and how they make reward hacking happen. To investigate it, we sample inpainting examples in Figure 3 and Figure 4. We report the following findings and insights.

**Reward models exhibit biases in brightness, composition, and color scheme.** As evidenced by the results from HPSv2 and PickScore—the second and third images in each sub-figure of Figure 3 respectively, we observe notable biases in their preferences. HPSv2 tends to favor images with bright lighting, complex composition with rich details, and vivid colors. In contrast, PickScore shows a preference for dim lighting, simple composition with few details, and muted colors.

**Biases in reward models affect different baseline models in distinct ways.** Although each reward model has its own inherent biases, we find that their influence varies across baseline models. For instance, BrushNet trained using HPSv2 produces inpainting outputs characterized by excessively bright lighting, overly intricate details, and unnaturally vivid colors—they seem deviate from human aesthetic preferences. In contrast, FLUX.1 Fill trained using HPSv2 generates visually pleasing results. PickScore shows a similar disparity in performance. It stems from the characteristics of baseline models as shown in Figure 4: BrushNet generates vibrant images, making PickScore particularly suitable for it; while FLUX.1 Fill produces plain images, aligning with HPSv2's property.

**Ensemble shows generality and generalization by mitigating biases.** Ensemble, a simple and straightforward method implemented through reward ensembling, exhibits strong versatility across models by producing balanced and aesthetically pleasing inpainting results, as shown in Figure 3 and Figure 4. It likely stems from Ensemble's ability to mitigate biases inherent in reward models.

## 7    ABLATION STUDIES AND COMPARISONS WITH STATE-OF-THE-ART

We name our methods **BruPA** and **FluPA**—BrushNet and FLUX.1 Fill with Ensemble-based preference alignment. We compare them with state-of-the-art image inpainting models. Specifically, the following methods are compared (they are introduced in section 2): SDI (Rombach et al., 2022), CNI (Zhang et al., 2023b), BLD (Avrahami et al., 2023), PowerPoint (Zhuang et al., 2024),

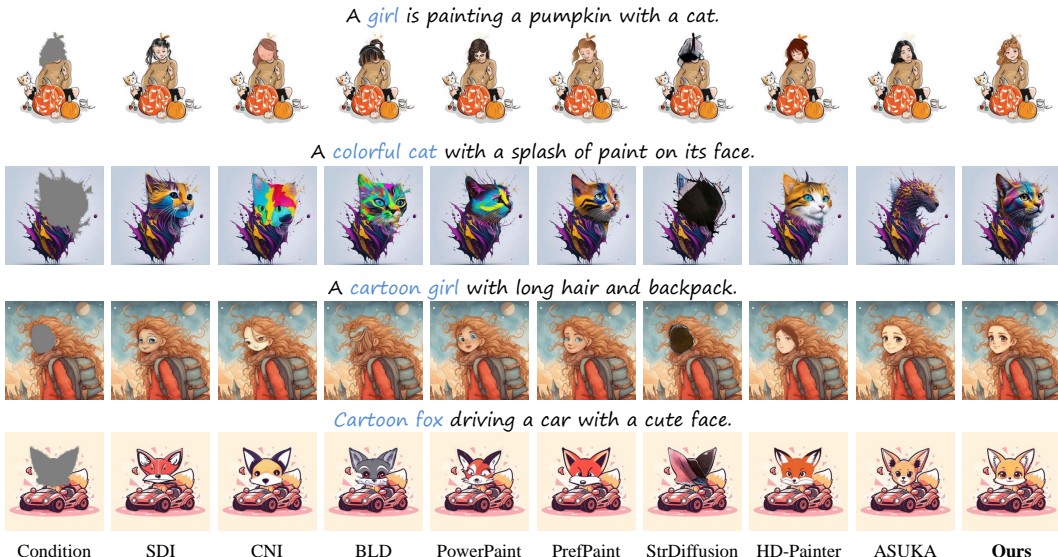

Figure 5: Qualitative comparisons with state-of-the-art image inpainting models.

Figure 6: **User studies.** We compare each pair of models by randomly sampling 100 pairs from their inpainting results. We invite 30 volunteers to participate in a blind assessment to determine which one is better ("A win", "B win", or "tie") based on their preferences. We report the **winning rates**.

BrushNet (Ju et al., 2024), PrefPaint (Liu et al., 2024b), StrDiffusion (Liu et al., 2024a), FLUX.1 Fill (BlackForestLabs, 2024b), HD-Painter (Manukyan et al., 2023), and ASUKA (Wang et al., 2025c), where *BrushNet and FLUX.1 Fill are also the baseline models for ablation studies*.

**Ablation studies.** We report quantitative and qualitative ablation studies, before and after preference alignment training, in Table 4 and Figure 4, respectively. After preference alignment using Ensemble, our method significantly surpasses the baseline models by achieving much better results and yielding visually appealing results. We further conduct ablation studies on a new dataset, reported in Table 3. It also confirms that our improvement is generalizable across data distributions.

**Comparisons with state-of-the-art.** Table 4 reports the results. Our BruPA and FluPA set new state-of-the-arts, attaining the best results on all evaluations and the second-best results in nearly half of the cases. Notably, even on coarser metrics—CLIPScore, VQAScore, and Perception (analyzed in section 4)—our methods still outperform competitors. Besides, BruPA and FluPA significantly outperform BrushNet and FLUX.1 Fill, i.e, the baselines before applying preference alignment. The qualitative results are reported in Figure 5, and our model generates images with better aesthetics.

**User studies.** As shown in Figure 6, our models align with human preferences better.

## 8 CONCLUSION AND LIMITATION DISCUSSIONS

We conduct extensive studies on image inpainting with preference alignment and obtain key insights into the effectiveness, scalability, and challenges in achieving alignment. We find that a simple ensemble method mitigate biases and achieve non-trivial results. Yet, our work is confined to images and DPO training; future extensions can generalize these findings to video, 3D data, and RLHF.

**Reproducibility statement.** We conduct experiments using the official implementations of Brush-Net (Ju et al., 2024) and FLUX.1 Fill (BlackForestLabs, 2024b). The DPO loss is implemented based on the official code from Wallace et al. (2024). Additional implementation details are also provided in section 4, section 5, and the Appendix. Our code will be open-sourced to ensure reproducibility.

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

## A    DETAILS TO MAKE GPT-4 AS AN EVALUATOR

Given GPT-4's (Achiam et al., 2023) strong multi-modal understanding capabilities, we use it to evaluate image inpainting results. Specifically, we provide GPT-4 with (1) a system prompt, (2) an input image, (3) the mask to inpaint, (4) an inpainting prompt, (5) and the inpainting result.

We randomly selected 500 inpainting pairs and invited volunteers to judge which result is better. Our human raters were drawn completely at random from a large and diverse volunteer pool. This random selection ensures that the evaluation is representative. To quantify agreement, we compute the alignment accuracy between the reward models and human raters, defined as the average proportion of cases in which both the raters and the model prefer the same result. In our evaluation, GPT-4 achieves 86% accuracy; HPSv2: 82%, PickScore: 80%, ImageReward: 77%, and Aesthetic: 80%. We acknowledge that, due to the complexity of real-world scenarios, the raters we are able to recruit may still exhibit minor biases.

To eliminate the potential biases in our design of prompts, we provide more results on the weights of the criteria—aesthetic quality, structural coherence, and semantic alignment, in Table 5. Our BruPA and FluPA consistently outperform the baseline models by a large margin, showing the significant and robust improvement of our method in GPT-4 evaluations.

Table 5: GPT-4 evaluation results with different weight combinations: [aesthetic quality]-[structural coherence]-[semantic alignment]. "auto" represents the weight is determined by GPT-4 itself.

| inpainting model | 40-30-30 | 30-40-30 | 30-30-40 | 33-33-33 | auto |
|---|---|---|---|---|---|
| BrushNet | 79.391 | 78.423 | 77.937 | 78.169 | 77.388 |
| BruPA | **83.054** | **82.538** | **82.388** | **82.226** | **81.036** |
| FLUX.1 Fill | 83.935 | 83.746 | 83.229 | 83.217 | 81.649 |
| FluPA | **87.609** | **87.559** | **87.368** | **87.243** | **85.309** |

Here, to further evaluate the results, we include an open source vision-language foundation model—Qwen3-VL (QwenTeam, 2025)—for evaluation. The results with means and standard deviations are reported in Table 6. Evaluation results of GPT-4 and Qwen3-VL are broadly stable, with our BruPA and FluPA delivering consistently remarkable performance gains.

Table 6: Evaluation by GPT-4 and Qwen3-VL

| inpainting model | GPT-4 | | Qwen3-VL | |
|---|---|---|---|---|
| | Brush. | Edit. | Brush. | Edit. |
| SDI | $79.004 \pm 0.203$ | $63.921 \pm 0.410$ | $90.323 \pm 0.245$ | $88.300 \pm 0.310$ |
| CNI | $74.173 \pm 0.817$ | $60.751 \pm 0.590$ | $88.101 \pm 0.277$ | $85.519 \pm 0.337$ |
| BLD | $71.794 \pm 0.657$ | $62.690 \pm 0.736$ | $85.719 \pm 0.461$ | $86.886 \pm 0.336$ |
| PowerPoint | $78.241 \pm 0.912$ | $63.092 \pm 0.738$ | $89.763 \pm 0.250$ | $85.682 \pm 0.425$ |
| PrefPaint | $80.327 \pm 0.293$ | $60.815 \pm 0.392$ | $91.249 \pm 0.220$ | $83.269 \pm 0.449$ |
| StrDiffusion | $34.255 \pm 0.159$ | $25.200 \pm 0.236$ | $49.560 \pm 0.394$ | $40.867 \pm 0.336$ |
| HD-Painter | $\underline{85.016} \pm 0.626$ | $\underline{69.089} \pm 0.628$ | $92.361 \pm 0.026$ | $87.389 \pm 0.325$ |
| ASUKA | $75.140 \pm 1.022$ | $58.686 \pm 0.477$ | $84.597 \pm 0.458$ | $73.614 \pm 0.980$ |
| BrushNet | $79.391 \pm 0.375$ | $57.046 \pm 1.151$ | $89.930 \pm 0.396$ | $85.064 \pm 0.450$ |
| **BruPA (ours)** | $83.054 \pm 0.886$ | $61.081 \pm 0.764$ | $92.255 \pm 0.163$ | $\underline{88.686} \pm 0.248$ |
| FLUX.1 Fill | $83.935 \pm 0.414$ | $67.465 \pm 0.463$ | $\underline{92.752} \pm 0.145$ | $88.649 \pm 0.306$ |
| **FluPA (ours)** | $\mathbf{87.609} \pm 0.291$ | $\mathbf{72.307} \pm 0.560$ | $\mathbf{94.501} \pm 0.154$ | $\mathbf{91.603} \pm 0.220$ |

The system prompt designed by us is given below:

> **GPT-4 System Prompt for Image Inpainting Evaluation**
>
> You are a human expert in analysis of image inpainting. Please evaluate the image inpainting result based on the following three criteria:
>
> - Aesthetic Quality (0–40 points):
>   - Visual appeal in color harmony, composition, style coherence
>   - Texture realism and naturalness
> - Structural Coherence (0–30 points)
>   - Preservation of geometric structures and content continuity
>   - Seamlessness at mask boundaries
> - Semantic Alignment (0–30 points)
>   - Faithfulness to the Text Prompt instructions
>   - Contextual consistency of added or restored content
>
> For each criterion, provide:
>
> - A sub-score.
> - A 1–2-sentence justification.
>
> Then compute the total score (0–100).

## B    SEARCHES OF HYPER-PARAMETERS

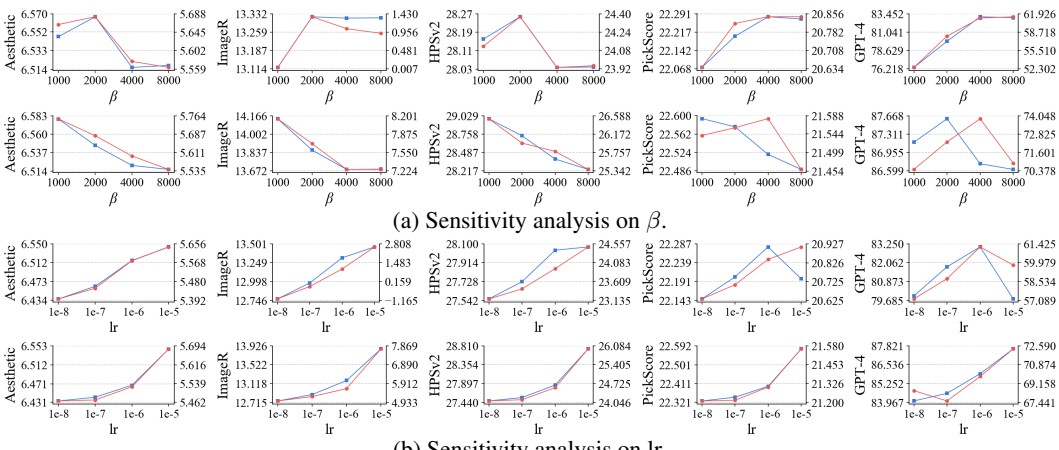

Figure 7: **Searches** of $\beta$, i.e., sub-figure (a), and learning rate (lr), i.e., sub-figure (b). The first and second row of each sub-figure is based on **BrushNet** and **FLUX.1 Fill**, respectively.

We conduct hyper-parameter searches for Ensemble, as shown in Figure 7. For Ensemble, we finally adopt a learning rate of 1e-6, and set $\beta = 4000$ for BrushNet; while using a learning rate of 1e-5, and set $\beta = 2000$ for FLUX.1 Fill. For HPSv2, we use a learning rate of 1e-5 with $\beta = 4000$ for BrushNet; and a learning rate of 1e-6 with $\beta = 8000$ on FLUX.1 Fill. For PickScore, we set the learning rate to 1e-7 and use $\beta = 2000$ for both BrushNet and FLUX.1 Fill.

We also find that fine-tuning these hyper-parameters is not critical. Even when using fixed moderate value ($\beta = 4000$ and lr $= 1e-6$, as suggested by Wallace et al. (2024)) across all experiments, our method still far exceeds the baseline w/o DPO: BruPA on BrushBench (83.133 vs. 79.391), BruPA on EditBench (72.442 vs. 57.046), FluPA on BrushBench (86.337 vs. 83.935), FluPA on EditBench (77.908 vs. 66.979). The new results remain the SOTA by outperforming previous best method: BrushBench (86.337 vs. 85.016) and EditBench (77.908 vs. 69.087). Besides, Ensemble is less sensitive to the change of hyper-parameter variations than single reward models like HPSv2 and PickScore. We hypothesis that this robustness aries because the Ensemble integrates different reward models, thereby mitigating the sensitivity inherent to individual reward model.

## C    DETAILS OF OUR ENSEMBLE METHOD

**Definition of our Ensemble:** For a set of images generated with different seeds but the same prompt, first rank all images by the score of each reward model and record their ranks. Then average the ranks of a single image across all reward models to obtain its Ensemble rank. The image with the highest Ensemble rank is selected as the winning sample, and the one with the lowest rank is selected as the losing sample. In this way, the preferences from different reward models are effectively integrated.

---

**Algorithm 1:** Ensemble Method

---

**Input** : $\mathcal{I}$: set of images generated under different seeds but the same prompt;
$\qquad$ $\mathcal{M}$: set of metrics;
$\qquad$ $s[m][i]$: score of metric $m \in \mathcal{M}$ for image $i \in \mathcal{I}$
**Output:** Ensemble$[i]$ for each $i \in \mathcal{I}$

1 **begin**
2 $\quad$ **for** *metric $m \in \mathcal{M}$* **do**
3 $\qquad$ **for** *image $i \in \mathcal{I}$* **do**
4 $\qquad\quad$ | $\quad$ rank$[m][i] \leftarrow \arg \mathrm{sort}\, s[m][i]$ ;
5 $\qquad$ **end for**
6 $\quad$ **end for**
7 $\quad$ **for** *image $i \in \mathcal{I}$* **do**
8 $\qquad$ **for** *metric $m \in \mathcal{M}$* **do**
9 $\qquad\quad$ | $\quad$ Ensemble$[i] \leftarrow \sum_{m \in \mathcal{M}} \mathrm{rank}[m][i]/|\mathcal{M}|$ ;
10 $\qquad$ **end for**
11 $\quad$ **end for**
12 $\quad$ Winning sample $\leftarrow \arg \min_{i \in \mathcal{I}}$ Ensemble$[i]$ ;
13 $\quad$ Losing sample $\leftarrow \arg \max_{i \in \mathcal{I}}$ Ensemble$[i]$ ;
14 **end**

---

## D    DETAILS OF VLM-BASED EVALUATION

We report the details of VLM-based evaluation, i.e., GPT-4 and Qwen3-VL, in Table 7. According to both GPT-4 and Qwen3-VL, our BruPA consistently improves image quality across all aspects—aesthetic appeal, structure accuracy, and semantic alignment.

Table 7: Details of VLM-based evaluations.

(a) GPT-4

| inpainting model | overall | aesthetics | structure | semantics |
|---|---|---|---|---|
| BrushNet | 79.391 | 30.301 | 24.035 | 25.055 |
| BruPA | **83.054** | **33.516** | **24.069** | **25.469** |
| FluPA | 83.935 | 32.617 | 24.997 | 26.321 |
| FLUX.1 Fill | **87.609** | **35.100** | **25.627** | **26.883** |

(b) Qwen3-VL

| inpainting model | overall | aesthetics | structure | semantics |
|---|---|---|---|---|
| BrushNet | 89.930 | 35.141 | 27.395 | 27.393 |
| BruPA | **92.255** | **36.032** | **28.018** | **28.204** |
| FLUX.1 Fill | 92.752 | 36.244 | 28.080 | 28.426 |
| FluPA | **94.501** | **36.972** | **28.477** | **29.052** |

## E    COMPUTE COSTS OF EACH REWARD MODEL

We report the compute cost of each reward models for giving score to a single sample in Table 8. Note that our Ensemble method only introduces additional cost for scoring samples and constructing preference pairs; the cost for training remains the same as in standard methods.

Table 8: Compute cost for giving score to a single sample.

| reward model | CLIPScore | Aesthetic | ImageR | PickScore | HPSv2 | VQAScore | UnifiedR | Perception | HPSv3 |
|---|---|---|---|---|---|---|---|---|---|
| cost (seconds) | 0.154 | 0.048 | 0.0759 | 0.0319 | 0.219 | 0.288 | 3.051 | 0.155 | 0.379 |

# F MORE RESULTS ON REWARD MODEL BIAS STUDIES

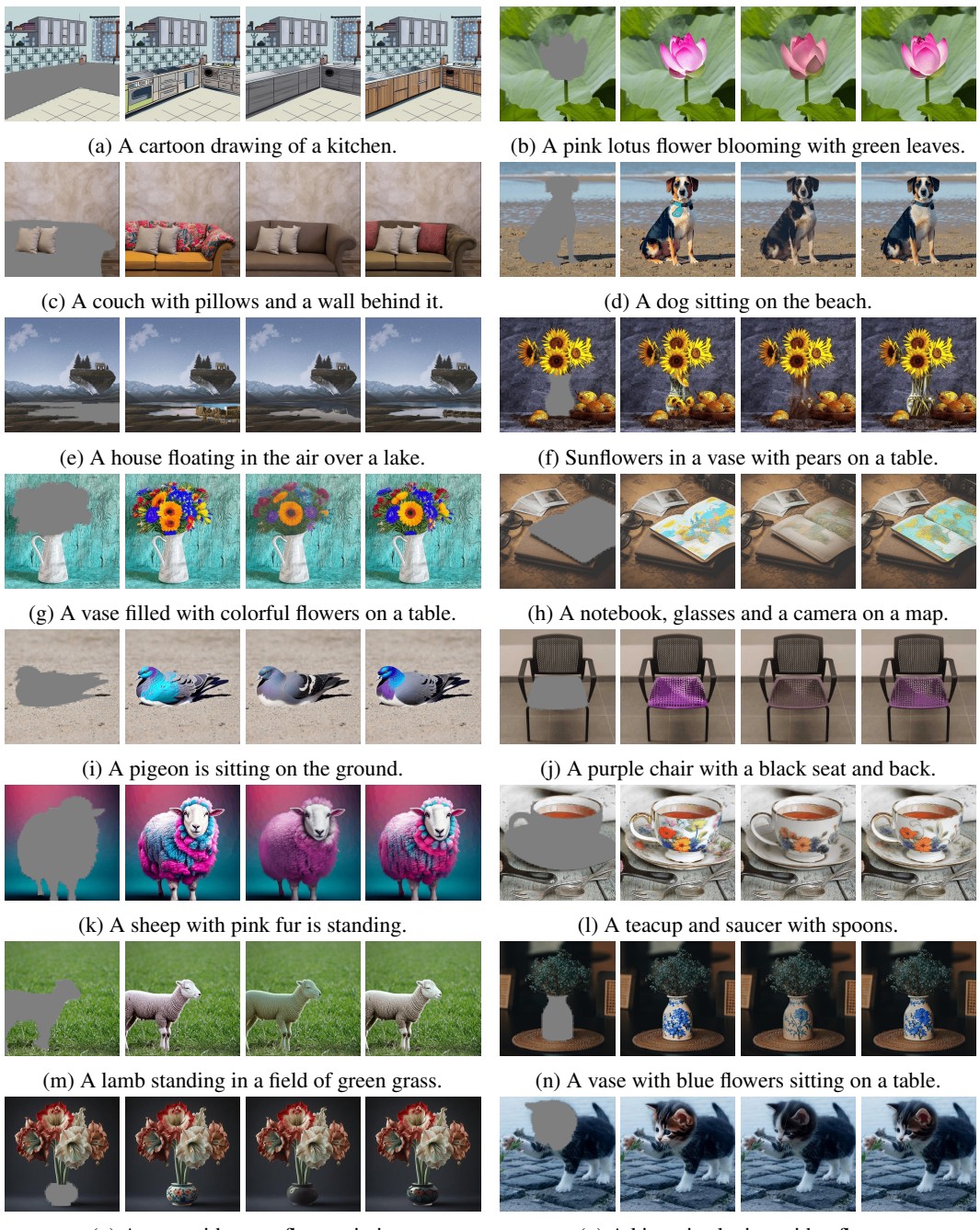

(a) A cartoon drawing of a kitchen.

(b) A pink lotus flower blooming with green leaves.

(c) A couch with pillows and a wall behind it.

(d) A dog sitting on the beach.

(e) A house floating in the air over a lake.

(f) Sunflowers in a vase with pears on a table.

(g) A vase filled with colorful flowers on a table.

(h) A notebook, glasses and a camera on a map.

(i) A pigeon is sitting on the ground.

(j) A purple chair with a black seat and back.

(k) A sheep with pink fur is standing.

(l) A teacup and saucer with spoons.

(m) A lamb standing in a field of green grass.

(n) A vase with blue flowers sitting on a table.

(o) A vase with some flowers in it.

(p) A kitten is playing with a flower.

Figure 8: **More results on reward model bias studies using BrushNet.** In each sub-figure, the four images (from left to right) display: the *masked image*, followed by inpainting results from models trained using *HPSv2*, *PickScore*, and *Ensemble*. For optimal detail, view figures zoomed in.

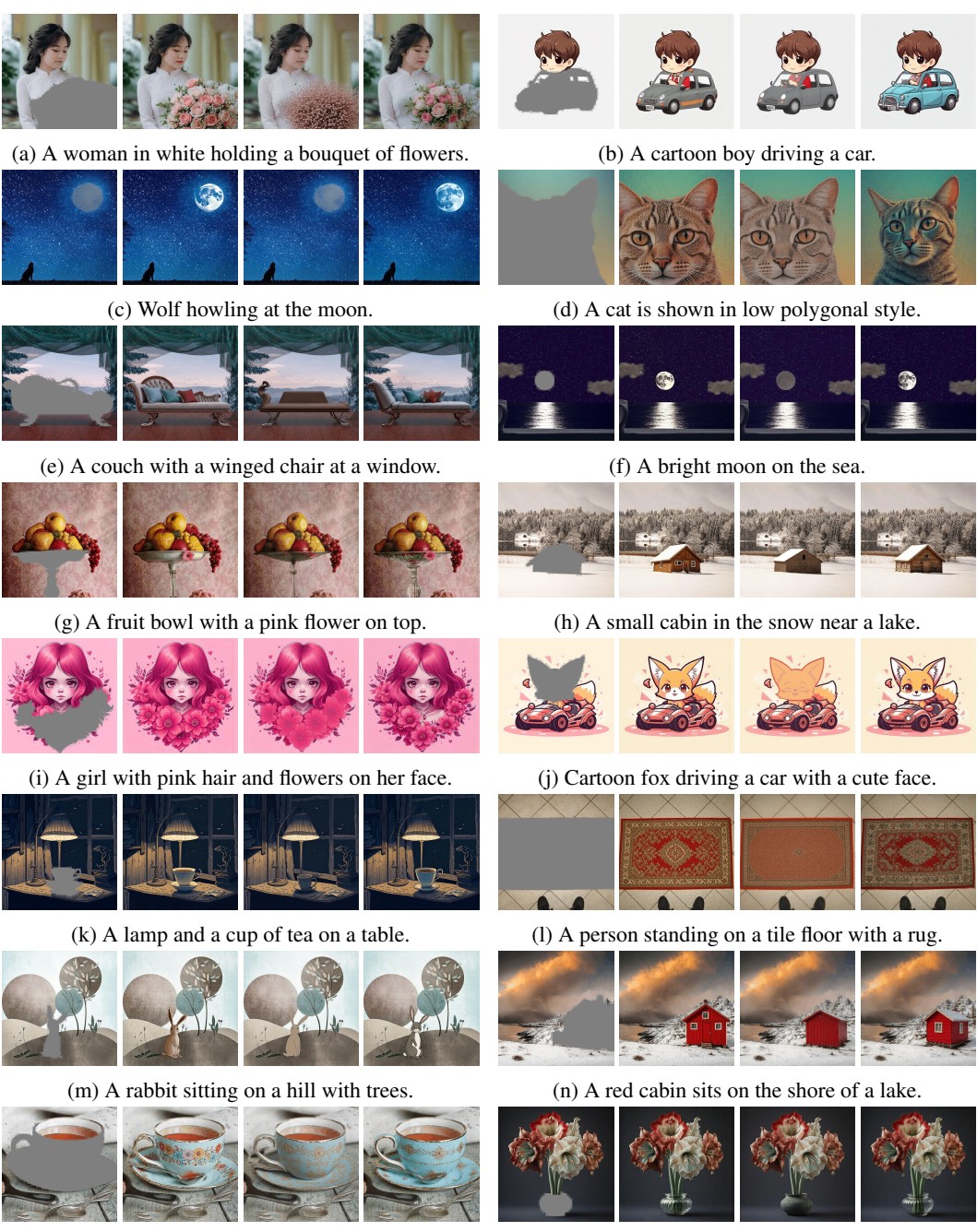

(a) A woman in white holding a bouquet of flowers.

(b) A cartoon boy driving a car.

(c) Wolf howling at the moon.

(d) A cat is shown in low polygonal style.

(e) A couch with a winged chair at a window.

(f) A bright moon on the sea.

(g) A fruit bowl with a pink flower on top.

(h) A small cabin in the snow near a lake.

(i) A girl with pink hair and flowers on her face.

(j) Cartoon fox driving a car with a cute face.

(k) A lamp and a cup of tea on a table.

(l) A person standing on a tile floor with a rug.

(m) A rabbit sitting on a hill with trees.

(n) A red cabin sits on the shore of a lake.

(o) A teacup and saucer with spoons.

(p) A vase with flowers in it on a dark background.

Figure 9: **More results on reward model bias studies using FLUX.1 Fill.** In each sub-figure, the four images (from left to right) display: the *masked image*, followed by inpainting results from models trained using *HPSv2*, *PickScore*, and *Ensemble*. For optimal detail, view figures zoomed in.

