# OpenReview forum: "Follow-Your-Preference: Towards Preference-Aligned Image Inpainting"
_ICLR.cc/2026/Conference — ICLR 2026 Poster_

### Official Review · Reviewer_9M76 · 2025-10-26

**Soundness:** 3
**Presentation:** 2
**Contribution:** 2
**Rating:** 6
**Confidence:** 4

**Summary:**

The paper studies image inpainting with preference alignment by revisiting basic questions rather than proposing a new architecture. Using Direct Preference Optimization, the authors construct preference datasets from nine public reward models and explore (i) which reward models yield useful training signals, (ii) how preference data scale with more candidates and samples, (iii) how reward hacking emerges, and (iv) whether a simple ensemble of reward models mitigates biases. Experiments span two inpainting backbones (BrushNet and FLUX.1 Fill) that cover diffusion and flow matching, evaluated on BrushBench and EditBench. Results show improvements across automatic metrics, GPT‑4 scoring, and human studies, without architectural changes or new datasets.

The paper offers a careful, evidence‑driven empirical study of preference alignment for inpainting, including scaling analyses and bias diagnostics. The ensemble strategy consistently improves or matches the best reward models across backbones and benchmarks. However, the evaluation pipeline relies heavily on GPT‑4 as a “fair” judge with limited validation and reporting details. The ensemble design and computational cost are underspecified. The bias analysis is qualitative mainly and could benefit from quantitative measurements.

**Strengths:**

- Evaluates nine reward models and two distinct backbones: BrushNet (diffusion) and FLUX.1 Fill (flow matching), across two benchmarks. This breadth supports generalizability claims.
- The paper concisely derives the DPO loss specialized to visual generation, supporting technical soundness.
- The paper documents brightness/composition/color‑scheme biases, e.g., HPSv2 favoring bright, detailed, vivid images; PickScore favoring dimmer, simpler, muted ones, providing novel qualitative insight.

**Weaknesses:**

- GPT‑4 is positioned as a “fair” evaluator with 86% accuracy vs. volunteers, but rater selection, agreement metrics, and sampling protocol are under‑reported. This affects evaluation validity. The evaluation prompt fixes specific criteria and weights (0–100), which may embed subjective biases. This limits generality. Details such as inter‑rater reliability, demographic diversity, and task difficulty balance are not provided. No direct evidence was found in the manuscript. This reduces statistical rigor.
- Ensemble is defined as average ranking of all reward models with no exploration of weighting schemes or subset selection, limiting technical transparency.
- The paper omits compute/latency costs for scoring with nine reward models and generating 16 candidates per prompt. No direct evidence found in the manuscript.
- Bias findings (brightness/composition/color) rely on visual inspection without quantitative measures, limiting scientific rigor.

**Questions:**

- The paper exclusively relies on GPT-4 as the “fair evaluator,” reporting 86% accuracy against volunteer judgments, yet it omits any cross-validation with other Vision-Language Models (VLMs) such as GPT-4o, Gemini, or Qwen2.5-VL, which are widely available and architecturally diverse.
- The “Ensemble” method is merely the average ranking of all reward models, without exploring learned or heuristic integration schemes, or even VLM-based approaches.

---

> ### Author Response · Authors · 2025-11-23
>
> Thank you for your insightful comments. We have revised our paper carefully and highlighted the updates using blue color in the new manuscript. Please find our response below.
>
> > **Weakness 1 & Question 1:** Reliability of GPT-4 as an evaluator
>
> Thank you for raising this concern.
>
> First, we have added more details about the evaluation of GPT-4 in **Appendix A**. Specifically, we randomly selected 500 inpainting pairs and invited volunteers to judge which result is better. Our human raters were drawn completely at random from a large and diverse volunteer pool. This random selection ensures that the evaluation is representative. To quantify agreement, we compute the alignment accuracy between the reward models and human raters, defined as the average proportion of cases in which both the raters and the model prefer the same result. We acknowledge that, due to the complexity of real-world scenarios, the raters we are able to recruit may still exhibit minor biases. We will explicitly include this limitation in the final version’s Limitation section.
>
> Next, to eliminate the potential biases in our design of prompts, we provide more results of different weight combinations of the criteria—aesthetic quality, structural coherence, and semantic alignment, in **Table 5**, given below. Our BruPA and FluPA consistently outperform the baseline models by a large margin, showing the significant and robust improvement of our method in GPT-4 evaluations.
>
> **Table 5: GPT-4 evaluation results with different weight combinations: [aesthetic quality]-[structural coherence]-[semantic alignment]. “auto” represents the weight is determined by GPT-4 itself.**
>
> | Inpainting model | 40-30-30 | 30-40-30 | 30-30-40 | 33-33-33 | auto   |
> |------------------|----------|----------|----------|----------|--------|
> | BrushNet         | 79.391   | 78.423   | 77.937   | 78.169   | 77.388 |
> | **BruPA**        | **83.054** | **82.538** | **82.388** | **82.226** | **81.036** |
> | FLUX.1 Fill      | 83.935   | 83.746   | 83.229   | 83.217   | 81.649 |
> | **FluPA**        | **87.609** | **87.559** | **87.368** | **87.243** | **85.309** |
>
>
> Finally, to further evaluate the results, we include an open-source vision-language foundation model—Qwen3-VL—for evaluation. The results with means and standard deviations are reported in **Table 6**, given below. Evaluation results of GPT-4 and Qwen3-VL are broadly stable, with our BruPA and FluPA delivering consistently remarkable performance gains.
>
> **Table 6: Evaluation by GPT-4 and Qwen3-VL.**
>
> | Inpainting model | GPT-4 Brush.       | GPT-4 Edit.        | Qwen3-VL Brush.      | Qwen3-VL Edit.       |
> |------------------|--------------------|--------------------|----------------------|----------------------|
> | SDI              | 79.004 ± 0.203     | 63.921 ± 0.410     | 90.323 ± 0.245       | 88.300 ± 0.310       |
> | CNI              | 74.173 ± 0.817     | 60.751 ± 0.590     | 88.101 ± 0.277       | 85.519 ± 0.337       |
> | BLD              | 71.794 ± 0.657     | 62.690 ± 0.736     | 85.719 ± 0.461       | 86.886 ± 0.336       |
> | PowerPoint       | 78.241 ± 0.912     | 63.092 ± 0.738     | 89.763 ± 0.250       | 85.682 ± 0.425       |
> | PrefPaint        | 80.327 ± 0.293     | 60.815 ± 0.392     | 91.249 ± 0.220       | 83.269 ± 0.449       |
> | StrDiffusion     | 34.255 ± 0.159     | 25.200 ± 0.236     | 49.560 ± 0.394       | 40.867 ± 0.336       |
> | HD-Painter       | 85.016 ± 0.626 | 69.089 ± 0.628 | 92.361 ± 0.026       | 87.389 ± 0.325       |
> | ASUKA            | 75.140 ± 1.022     | 58.686 ± 0.477     | 84.597 ± 0.458       | 73.614 ± 0.980       |
> | BrushNet         | 79.391 ± 0.375     | 57.046 ± 1.151     | 89.930 ± 0.396       | 85.064 ± 0.450       |
> | **BruPA (ours)** | 83.054 ± 0.886     | 61.081 ± 0.764     | 92.255 ± 0.163       | 88.686 ± 0.248 |
> | FLUX.1 Fill      | 83.935 ± 0.414     | 67.465 ± 0.463     | 92.752 ± 0.145 | 88.649 ± 0.306       |
> | **FluPA (ours)** | **87.609 ± 0.291** | **72.307 ± 0.560** | **94.501 ± 0.154**   | **91.603 ± 0.220**   |

---

> ### Author Response · Authors · 2025-11-23
>
> > **Weakness 2 & Question 2:** More exploration of Ensemble
>
> Thank you for pointing it out.
>
> We further report results for two new variants—**Linear Weighted Ensemble** (named **BruPA (linear) and FluPA (linear)**) and **Softmax Weighted Ensemble** (named **BruPA (softmax)** and **FluPA (softmax)**)—in **Appendix C.2**. We report the results in **Table 7**, also given below. In our original BruPA (vanilla) and FluPA (vanilla), each reward model is given equal weight. In comparison, in BruPA (linear) and FluPA (linear), each reward model is weighted by its corresponding GPT-4 score (from Table 1 and Table 2), granting greater influence to higher-performing models. The BruPA (softmax) and FluPA (softmax) further normalize these weights via a softmax transformation. Based on GPT-4 evaluations, both linear and softmax variants outperform the vanilla Ensemble approach.
>
> **Table 7: Comparisons of different Ensemble variants.**
>
> | Variant           | CLIPScore        | Aesthetic        | ImageR            | PickScore         | HPSv2            | VQAScore          | UnifiedR          | Perception        | HPSv3            | GPT-4           |
> |-------------------|------------------|------------------|-------------------|-------------------|------------------|-------------------|-------------------|-------------------|------------------|-----------------|
> | BruPA (vanilla)   | 26.547           | 6.516            | 13.315            | **22.279**        | 28.037           | 9.093             | 3.371             | 26.390            | 6.276            | 83.054          |
> | BruPA (linear)    | 26.568           | **6.521**        | **13.342**        | 22.274            | 28.050           | **9.098**         | **3.372**         | 26.400            | 6.288            | **83.479**      |
> | BruPA (softmax)   | **26.573**       | 6.516           | 13.136            | 22.265            | **28.073**       | 9.092             | 3.359             | **26.409**        | **6.310**        | 83.136     |
> | FluPA (vanilla)   | 26.436           | 6.546   | 13.859            | 22.577            | 28.735           | 9.152             | **3.457**         | 26.096            | 7.000            | 87.609  |
> | FluPA (linear)    | **26.513**       | **6.556**        | **14.030**        | **22.605**        | **28.815**       | **9.172**         | 3.452             | **26.171**        | **7.136**        | **89.190**      |
> | FluPA (softmax)   | 26.434           | 6.553            | 13.834            | 22.538            | 28.609           | 9.166             | 3.443             | 26.080            | 6.984            | 88.145          |
>
> As established in Section 4, VQAScore fails to serve as either a reliable evaluator or a valid reward signal provider. To address this limitation, we propose a new baseline by eliminating this underperforming reward model from the Ensemble framework. We evaluate this modified setup on BrushBench, with results summarized in **Appendix C.3** and reported in **Table 8**. Notably, both BrushPA and FluPA exhibit substantial improvements in GPT-4 evaluation scores following the exclusion of VQAScore. This finding offers preliminary evidence that removing weaker reward models—despite reducing the diversity of reward signals—can yield enhanced preference alignment outcomes. A key insight emerges: when weaker models are included in the ensemble, a trade-off arises between the diversity of reward signals and their overall effectiveness. This observation paves the way for several promising research avenues, which we aim to explore in future investigations.
>
> **Table 8: Ablations on VQAScore.**
>
> | Inpainting model      | CLIPScore       | Aesthetic       | ImageR          | PickScore        | HPSv2           | VQAScore         | UnifiedR         | Perception       | HPSv3           | GPT-4            |
> |-----------------------|-----------------|-----------------|-----------------|------------------|-----------------|------------------|------------------|------------------|-----------------|------------------|
> | BruPA                 | 26.547          | 6.516           | 13.315          | **22.279**       | 28.037          | **9.093**        | **3.371**        | **26.390**       | 6.276           | 83.054           |
> | BruPA w/o VQAScore    | **26.555**      | **6.524**       | **13.337**      | 22.278           | **28.055**      | 9.082            | 3.362            | 26.385           | **6.306**       | **83.659**       |
> | FluPA                 | 26.436          | 6.546     | 13.859          | 22.577           | 28.735          | 9.152            | 3.457            | 26.096           | 7.000           | 87.609           |
> | FluPA w/o VQAScore    | **26.438**      | **6.576**       | **14.017**      | **22.605**       | **28.747**      | **9.186**        | **3.466**        | **26.113**       | **7.068**       | **88.972**       |

---

> ### Author Response · Authors · 2025-11-23
>
> > **Weakness 3:** Compute cost
>
> Thank you for pointing it out.
>
> We have reported the computation cost of each reward model in **Appendix G**. The results are presented in **Table 13**. Note that our Ensemble method only introduces additional cost during scoring samples for constructing preference pairs; the cost for training remains the same as in standard methods. We will also include a discussion of this point in the Limitation section of our final version.
>
> **Table 13: Compute cost for giving scores to a single sample.**
>
> | Reward model    | CLIPScore | Aesthetic | ImageR | PickScore | HPSv2 | VQAScore | UnifiedR | Perception | HPSv3 |
> |-----------------|-----------|-----------|--------|-----------|-------|----------|----------|------------|-------|
> | Cost (seconds)  | 0.154     | 0.048     | 0.0759 | 0.0319    | 0.219 | 0.288    | 3.051    | 0.155      | 0.379 |
>
> > **Weakness 4:** Quantitative results of bias
>
> Thank you for raising this concern.
>
> We have added the quantitative results in **Appendix D.1**. Because there is no such a model available for evaluating brightness, composition, and color, we therefore make an attempt to use vision-language foundation models—specifically GPT-4 and Qwen3-VL—for assessing generation biases. The system prompt is given in **Appendix D.1**, and the results are reported in **Table 9**. Quantitatively, the results strongly corroborate our qualitative analyses in section 6: HPSv2 exhibits a preference for images characterized by bright lighting, complex composition, and vivid colors; PickScore demonstrates the inverse tendency; and our Ensemble model occupies a middle ground between the two extremes.
>
> **Table 9: Quantitative results of biases evaluated by GPT-4 and Qwen3-VL.**
>
> ### (a) GPT-4
>
> | model        | reward model | brightness     | composition    | color          |
> |-------------|--------------|----------------|----------------|----------------|
> | BrushNet    | HPSv2        | **79.221**     | **82.979**     | **81.267**     |
> | BrushNet    | Ensemble     | 78.180 | 80.922 | 78.513 |
> | BrushNet    | PickScore    | 73.545         | 75.019         | 70.641         |
> | FLUX.1 Fill | HPSv2        | **76.617**     | **74.547**     | **70.978**     |
> | FLUX.1 Fill | Ensemble     | 76.412  | 74.057  | 70.496 |
> | FLUX.1 Fill | PickScore    | 76.124         | 73.512         | 69.824         |
>
> ### (b) Qwen3-VL
>
> | model        | reward model | brightness     | composition    | color          |
> |-------------|--------------|----------------|----------------|----------------|
> | BrushNet    | HPSv2        | **83.843**     | **88.310**     | **85.565**     |
> | BrushNet    | Ensemble     | 83.284 | 88.180 | 84.644  |
> | BrushNet    | PickScore    | 82.615         | 86.530         | 82.516         |
> | FLUX.1 Fill | HPSv2        | **82.750**     | **85.832**     | **81.695**     |
> | FLUX.1 Fill | Ensemble     | 82.527  | 85.611 | 81.454  |
> | FLUX.1 Fill | PickScore    | 82.508         | 85.481         | 81.389         |

---

> > ### Comment · Reviewer_9M76 · 2025-11-28
> >
> > Thanks for the response. Most of my concerns are addressed. I will keep my original score.

---

### Official Review · Reviewer_854V · 2025-10-26

**Soundness:** 4
**Presentation:** 4
**Contribution:** 3
**Rating:** 8
**Confidence:** 4

**Summary:**

This paper provides a comprehensive evaluation of preference alignment with multiple reward models for the task of image inpainting.
It provides useful insights into how different reward models affect the final performance of the model by studying biases in them that could cause reward hacking, and studying data scaling trends. Finally, it proposes that an ensemble of reward models can produce generalizable results and shows strong quantitative performance.

**Strengths:**

- The paper provides a thorough analysis of different reward models for image synthesis, such as 1) demonstrating how some reward models are not reliable, while valid reward models can still exhibit biases, 2) scalability of preference data, and 3) analysis on reward hacking,  through a systematic study. These can be useful analyses for the researchers working with preference alignment.
- It is the first work showing a systematic study of preference alignment with reward models for image synthesis.
- Experiment design is clear and well thought out, and the result is written clearly.
- This paper demonstrates a strong result by identifying reliable reward models and ensembling them, and this can be a baseline that the community could build upon.

**Weaknesses:**

The reviewer did not find significant weaknesses of the paper other than:
- Calibrated Multi-Preference Optimization for Aligning Diffusion Models (Lee et al., CVPR 2025) also considers a multi-reward formulation and could be compared against, but this work is only cited very briefly in the related work.

**Questions:**

It would be great if the rebuttal could include the comparison with the paper mentioned in the weaknesses section or a justification of why it cannot be included.

---

> ### Author Response · Authors · 2025-11-23
>
> Thank you for your positive comments. We have revised our paper carefully and highlighted the updates using blue color in the new manuscript. Please find our response below.
>
> > **Weakness 1:** Add a related work
>
> We appreciate the reviewer’s suggestion to incorporate this method as a baseline, and we agree that such a comparison would further strengthen the paper. Unfortunately, the authors of this work have not released their implementation, and given the limited rebuttal period, it is impractical for us to faithfully re-implement the method from scratch without risking inaccuracies. We would be happy to include this method in experiments once the official implementation becomes available. At this stage, we will include a **more detailed discussion of this method in the final version**. We write it below.
>
> Lee et al. introduce CaPO, a framework for aligning diffusion models using multiple reward models. It calibrates the reward scores from different reward models and optimizes the model by matching the calibrated rewards differences using a regression loss. In comparison, the Ensemble strategy introduced in our work is a simple, intuitive, and straightforward baseline that directly averages the rank of each sample across different reward models. Our intention is to provide a simple yet strong baseline that can serve future work in this emerging research direction.

---

> > ### Comment · Reviewer_854V · 2025-11-27
> >
> > I thank the authors for the rebuttal and I maintain my rating as accept.

---

### Official Review · Reviewer_XjVN · 2025-10-31

**Soundness:** 2
**Presentation:** 2
**Contribution:** 2
**Rating:** 4
**Confidence:** 3

**Summary:**

This paper conducts a systematic and extensive investigation into aligning image inpainting models with human preferences. Instead of proposing a novel architecture, the authors revisit fundamental questions about using preference alignment in this domain. They employ the DPO algorithm, leveraging nine different public, off-the-shelf reward models to automatically construct preference datasets without costly human annotation.

**Strengths:**

Pros:
1. It provides the community with valuable insights into the behavior of reward models and the dynamics of preference alignment.
2. The analysis and visualization of "reward hacking" (Figure 3) clearly demonstrates a crucial pitfall in naive preference alignment and provides a strong motivation for their proposed solution.
3. The proposed Ensemble method shows that significant gains can be achieved by thoughtfully combining existing tools rather than inventing complex new ones.

**Weaknesses:**

Cons:
1. The paper states the Ensemble is based on "the average ranking of all the reward models" but could be more explicit. Does this involve normalizing scores before averaging, or is it a direct average of rank positions? A more formal definition or algorithmic description would improve reproducibility.
2. In your analysis, some reward models (like VQAScore) were found to be unreliable evaluators. However, they seem to be included in the final Ensemble method. Did you test if an ensemble composed of only the most reliable/diverse reward models would perform better or worse than an ensemble of all nine? Is there a benefit to including weaker, but potentially diverse, signals?
3. More details on the Hyper-parameter Sensitivity are desired. The appendix shows searches for the DPO hyper-parameter β and the learning rate. How crucial was fine-tuning these parameters for achieving top performance? Is the Ensemble method more or less sensitive to these hyper-parameters compared to using a single reward model like HPSv2?
4. Does the DPO process primarily learn aesthetic qualities (e.g., better lighting, composition), or does it also improve semantic alignment to the text prompt? The results suggest an aesthetic improvement, but it would be interesting to know if prompt faithfulness also improved systematically.
5. The evaluation benchmarks (BrushBench, EditBench) primarily focus on object replacement, addition, or style modification. The paper does not explore how preference alignment performs on other critical inpainting tasks, such as: Object Removal: Where the primary goal is seamless and plausible background completion, not creative generation. Image Restoration: Such as removing scratches or blemishes, where fidelity and realism are paramount. It's unclear if aligning for "aesthetic preference" would be beneficial or even detrimental in these contexts, where the model should be as unobtrusive as possible.
6. The paper does identify biases (e.g., brightness, complexity). However, it stops short of deeply investigating why these biases exist. This is likely tied to the data on which the reward models were trained (e.g., HPSv2 was trained on a dataset of human-preferred images, which may have skewed towards professionally shot, well-lit photos). A more thorough discussion on the provenance of these biases would strengthen the paper's contribution and provide a clearer path toward creating less biased reward models in the future.

**Questions:**

See Weaknesses

---

> ### Author Response · Authors · 2025-11-23
>
> Thank you for your insightful comments. We have revised our paper carefully and highlighted the updates using blue color in the new manuscript. Please find our response below.
>
> > **Weakness 1:** Formal definition of Ensemble method
>
> Thank you for pointing this out.
>
> We have added a **detailed definition** and **algorithmic description** of our Ensemble method in **Appendix C.1**. Specifically, for a set of image candidates generated with different seeds but the same prompt, first rank all images by the score of each reward model and record their ranks. Then average the ranks of a single image across all reward models to obtain its Ensemble rank. The image with the highest Ensemble rank is selected as the winning sample, and the one with the lowest rank is selected as the losing sample. In this way, the preferences from different reward models are effectively integrated.
>
> > **Weakness 2:** Result without weak reward models like VQAScore
>
> Thank you for raising this very insightful point.
>
> We have included a new baseline in **Appendix C.3** by removing VQAScore from Ensemble. The results are reported in **Table 8**, given below. We observe a notable improvement in the GPT-4 evaluation scores for our BrushPA and FluPA after excluding VQAScore. This result provides preliminary evidence that removing weaker reward models—although it reduces the diversity of reward signals—can further improve preference alignment performance.
>
> Diversity and the effectiveness of reward signals seem to be a trade-off when weaker reward models are present. We believe this opens up several interesting research directions, and we look forward to exploring them in future work.
>
> **Table 8: Ablations on VQAScore.**
>
> | Inpainting model      | CLIPScore       | Aesthetic       | ImageR          | PickScore        | HPSv2           | VQAScore         | UnifiedR         | Perception       | HPSv3           | GPT-4            |
> |-----------------------|-----------------|-----------------|-----------------|------------------|-----------------|------------------|------------------|------------------|-----------------|------------------|
> | BruPA                 | 26.547          | 6.516           | 13.315          | **22.279**       | 28.037          | **9.093**        | **3.371**        | **26.390**       | 6.276           | 83.054           |
> | BruPA w/o VQAScore    | **26.555**      | **6.524**       | **13.337**      | 22.278           | **28.055**      | 9.082            | 3.362            | 26.385           | **6.306**       | **83.659**       |
> | FluPA                 | 26.436          | 6.546           | 13.859          | 22.577           | 28.735          | 9.152            | 3.457            | 26.096           | 7.000           | 87.609           |
> | FluPA w/o VQAScore    | **26.438**      | **6.576**       | **14.017**      | **22.605**       | **28.747**      | **9.186**        | **3.466**        | **26.113**       | **7.068**       | **88.972**       |
>
>
> > **Weakness 3:** Add explanation for the hyper-parameter sensitivity results
>
> We have expanded the discussion on hyper-parameter tuning in **Appendix B**.
>
> Specifically, in DPO, $\beta$ controls the strength of the KL regularization between the policy and the reference model. Larger $\beta$ emphasizes staying close to the reference model, while smaller $\beta$ allows for more aggressive preference-driven updates. Extreme settings—very small $\beta$ or very large $\beta$—can lead to over-optimization or under-adaptation, respectively. The learning rate determines the magnitude of policy updates: excessively large values cause instability, while smaller values stabilize training but may underfit the preference signal.
>
> Because we experiment with multiple reward models and policy models, we perform a light hyper-parameter search to ensure strong performance. However, we find that **fine-tuning these hyper-parameters is not critical**. Even when using **fixed moderate value** ($\beta=4000$ and $\mathrm{lr}=1e-6$, as suggested by [1]) **across all experiments**, our method still far exceeds the baseline w/o DPO: BruPA on BrushBench (83.054 vs. 79.391), BruPA on EditBench (61.186 vs. 57.046), FluPA on BrushBench (87.215 vs. 83.935), FluPA on EditBench (70.012 vs. 66.979). The new results remain the SOTA by outperforming the previous best method: BrushBench (87.215 vs. 85.016) and EditBench (70.012 vs. 69.087). Besides, Ensemble is less sensitive to the change of hyper-parameter variations than single reward models like HPSv2 and PickScore. We hypothesize that this robustness arises because the Ensemble integrates different reward models, thereby mitigating the sensitivity inherent to individual reward model.
>
> [1] Diffusion Model Alignment Using Direct Preference Optimization, CVPR 2024.

---

> ### Author Response · Authors · 2025-11-23
>
> > **Weakness 4:** Improvement on semantic alignment?
>
> We appreciate this insightful comment.
>
> We have added the details of VLM-based evaluations in **Appendix F**, and we report the results in **Table 12**, which is given below. To make the assessment more reliable and generalizable, we further include evaluations using Qwen3-VL. According to both GPT-4 and Qwen3-VL, our BruPA consistently improves image quality across all aspects—aesthetic quality, structure accuracy, and semantic alignment.
>
> **Table 12: Details of VLM-based evaluations.**
>
> **(a) GPT-4**
>
> | Inpainting model | Overall  | Aesthetics | Structure | Semantics |
> |------------------|----------|-----------|-----------|-----------|
> | BrushNet         | 79.391   | 30.301    | 24.035    | 25.055    |
> | **BruPA**        | **83.054** | **33.516** | **24.069** | **25.469** |
> | FluPA            | 83.935   | 32.617    | 24.997    | 26.321    |
> | **FLUX.1 Fill**  | **87.609** | **35.100** | **25.627** | **26.883** |
>
> **(b) Qwen3-VL**
>
> | Inpainting model | Overall  | Aesthetics | Structure | Semantics |
> |------------------|----------|-----------|-----------|-----------|
> | BrushNet         | 89.930   | 35.141    | 27.395    | 27.393    |
> | **BruPA**        | **92.255** | **36.032** | **28.018** | **28.204** |
> | FLUX.1 Fill      | 92.752   | 36.244    | 28.080    | 28.426    |
> | **FluPA**        | **94.501** | **36.972** | **28.477** | **29.052** |
>
>
> > **Weakness 5:** Other inpainting tasks
>
> Thank you for providing this valuable feedback.
>
> Following your suggestion, we make an attempt to extend our method to additional tasks like object removal, as demonstrated in **Appendix E**. For this new task, we build preference pairs and train our BruPA on the RORem dataset for 400 steps, following the same training recipe. We employ the RORD-Val validation dataset to evaluate the results, which are presented in **Table 11**, given below. The results demonstrate that our BruPA significantly outperforms the baseline model, BrushNet, on this new task. It indicates that our method can generalize to other image inpainting and editing tasks. We are excited to further explore the potential of our approach in future work.
>
> **Table 11: Results for object removal.**
>
> | Inpainting model | CLIPScore | PSNR   | PSNR-BG | LPIPS↓  |
> |------------------|-----------|--------|---------|--------|
> | BrushNet         | 87.640    | 0.116  | 43.445  | 0.082  |
> | **BruPA**        | **89.896**| **1.342** | **43.791** | **0.071** |
>
>
> > **Weakness 6:** Discussion on the provenance of reward model biases
>
> Thank you for this constructive suggestion.
>
> To better understand the sources of the biases in reward models, we analyze the training data of two representative reward models—HPSv2 and PickScore in **Appendix D.2**. Specifically, we sample 1000 preference pairs with 1000 winning samples and 1000 losing samples from HPDv2 and Pick-a-Pic, the training datasets used to train HPSv2 and PickScore, respectively. We then employ GPT-4 to evaluate these images along the attributes of brightness, composition, and color. The results are reported in **Table 10**, given below. As expected, HPDv2 exhibits **substantial differences** between the winning and losing samples across all three attributes. In contrast, Pick-a-Pic shows only **minor differences**. We believe these dataset characteristics explain why HPSv2 tends to favor bright, complex, and vivid images, whereas PickScore demonstrates the opposite behavior.
>
> **Table 10: Bias analysis on HPDv2 and Pick-a-Pic.**
>
> | dataset    | brightness win | brightness loss | brightness win−lose | composition win | composition loss | composition win−lose | color win        | color loss        | color win−lose    |
> |-----------|----------------|-----------------|---------------------|-----------------|------------------|----------------------|------------------|-------------------|-------------------|
> | HPDv2     | 40.641         | 39.236          | **1.405**           | 45.208          | 41.560           | **3.648**            | 42.340       | 38.269        | **4.071**         |
> | Pick-a-Pic| **42.421**     | **42.415**      | 0.006               | **46.144**      | **45.899**       | 0.245                | **43.943**       | **44.137**        | -0.194            |

---

### Official Review · Reviewer_mUgh · 2025-11-01

**Soundness:** 3
**Presentation:** 3
**Contribution:** 2
**Rating:** 4
**Confidence:** 3

**Summary:**

This paper investigates preference alignment for image inpainting, a relatively underexplored but increasingly important topic.
Rather than proposing a novel architecture, the authors systematically study how to align inpainting models with human aesthetic preferences through Direct Preference Optimization (DPO).

**Strengths:**

The paper is among the first to systematically analyze preference alignment in image inpainting, bridging the gap between text-to-image alignment studies and conditional image editing.

The experimental design is rigorous: nine reward models, multiple benchmarks, two architectures (diffusion and flow-based), and both automatic (GPT-4) and human evaluations.

The paper is well-written and easy to follow.

**Weaknesses:**

The paper shows limited algorithmic novelty, as the core approach mainly combines existing DPO techniques with a straightforward averaging ensemble.

The reliance on GPT-4 for evaluation may introduce instability and reproducibility concerns, as model-based judgments can vary over time and lack full transparency.

While the study highlights potential biases related to brightness, color, and composition, it lacks quantitative analysis or formal metrics to substantiate these observations.

**Questions:**

Did you try other ensemble strategies (e.g., weighted average or learned combination)?

---

> ### Author Response · Authors · 2025-11-23
>
> Thank you for your comments and constructive advice. We have revised our paper carefully and highlight the updates using blue color in the new manuscript. Please find our response below.
>
> > **Weakness 1:** Limited algorithmic novelty
>
> We would like to clarify our core contributions.
>
> As stated in the Abstract and Introduction, our goal is **not** to propose a novel algorithmic framework, but rather to go back to basics and revisit fundamental problems in preference-aligned image inpainting. In other words, the primary contribution of this work lies in providing **inspiring findings through systematic empirical studies**, providing important knowledge and establishing a **simple yet strong baseline** for this emerging research direction.
>
> To this end, we conduct extensive experiments across nine reward models, two benchmarks, and two baseline models with varying structures and generative algorithms. Our study reveals several key findings—for example, observable biases in reward models concerning brightness, composition, and color scheme, which can lead to reward hacking. Motivated by these observations, we introduce a simple yet effective strategy named Ensemble to mitigate such biases and achieve significant performance improvements, as demonstrated in experiments (e.g., Table 3, Table 4, Figure 5, and Figure 6).
>
> Meanwhile, to enrich the algorithmic novelty, as you also suggested in **Question 1**, we have included results of two new methods—**Linear Weighted Ensemble** and **Softmax Weighted Ensemble** in **Table 7** of **Appendix C.2**. In the Linear Weighted Ensemble (named **BruPA (linear)** and **FluPA (linear)**), each reward model is weighted by its corresponding GPT-4 score from Table 1 and Table 2, giving higher-performing models greater influence. The Softmax Weighted Ensemble (named **BruPA (softmax)** and **FluPA (softmax)**) further normalizes these weights using a softmax transformation. We observe that both Linear Weighted Ensemble and Softmax Weighted Ensemble outperform the vanilla Ensemble method based on GPT-4 evaluations. We hope such experimental results will enable future researchers to build upon our baseline findings for further algorithmic innovation and performance improvements.
>
> **Table 7: Comparisons of different Ensemble variants.**
>
> | Variant           | CLIPScore        | Aesthetic        | ImageR            | PickScore         | HPSv2            | VQAScore          | UnifiedR          | Perception        | HPSv3            | GPT-4           |
> |-------------------|------------------|------------------|-------------------|-------------------|------------------|-------------------|-------------------|-------------------|------------------|-----------------|
> | BruPA (vanilla)   | 26.547           | 6.516            | 13.315            | **22.279**        | 28.037           | 9.093             | 3.371             | 26.390            | 6.276            | 83.054          |
> | BruPA (linear)    | 26.568           | **6.521**        | **13.342**        | 22.274            | 28.050           | **9.098**         | **3.372**         | 26.400            | 6.288            | **83.479**      |
> | BruPA (softmax)   | **26.573**       | 6.516            | 13.136            | 22.265            | **28.073**       | 9.092             | 3.359             | **26.409**        | **6.310**        | 83.136          |
> | FluPA (vanilla)   | 26.436           | 6.546            | 13.859            | 22.577            | 28.735           | 9.152             | **3.457**         | 26.096            | 7.000            | 87.609          |
> | FluPA (linear)    | **26.513**       | **6.556**        | **14.030**        | **22.605**        | **28.815**       | **9.172**         | 3.452             | **26.171**        | **7.136**        | **89.190**      |
> | FluPA (softmax)   | 26.434           | 6.553            | 13.834            | 22.538            | 28.609           | 9.166             | 3.443             | 26.080            | 6.984            | 88.145          |

---

> ### Author Response · Authors · 2025-11-23
>
> > **Weakness 2:** Model-based judgements are unreliable
>
> Thank you for raising this concern. To address it, we have incorporated **Qwen3-VL** as an additional evaluation model, and the results are reported in **Table 6** of **Appendix A**, also given below. Qwen3-VL is an open-source vision-language foundation model, which improves both the transparency and reproducibility of our evaluation. Moreover, we include both the mean and standard deviation over three runs with different random seeds. It shows that the evaluation results from both GPT-4 and Qwen3-VL are generally stable and the performance gains of our BruPA and FluPA are consistently significant.
>
> We would also like to note that the majority of evaluation metrics used in contemporary image generation research are inherently **model-based**. Due to the high cost and limited scalability of human evaluation, fully avoiding model-based metrics is often impractical for large-scale experiments. To mitigate this, we use **10 evaluation metrics** for our experiments (such as Table 3 and Table 4), as well as **human studies** in Figure 6, to provide a more comprehensive assessment. We acknowledge that reliable and standardized evaluation for image generation remains an open challenge, and we hope future work will continue to develop more stable and transparent evaluation protocols.
>
> **Table 6: Evaluation by GPT-4 and Qwen3-VL.**
>
> | Inpainting model | GPT-4 Brush.       | GPT-4 Edit.        | Qwen3-VL Brush.      | Qwen3-VL Edit.       |
> |------------------|--------------------|--------------------|----------------------|----------------------|
> | SDI              | 79.004 ± 0.203     | 63.921 ± 0.410     | 90.323 ± 0.245       | 88.300 ± 0.310       |
> | CNI              | 74.173 ± 0.817     | 60.751 ± 0.590     | 88.101 ± 0.277       | 85.519 ± 0.337       |
> | BLD              | 71.794 ± 0.657     | 62.690 ± 0.736     | 85.719 ± 0.461       | 86.886 ± 0.336       |
> | PowerPoint       | 78.241 ± 0.912     | 63.092 ± 0.738     | 89.763 ± 0.250       | 85.682 ± 0.425       |
> | PrefPaint        | 80.327 ± 0.293     | 60.815 ± 0.392     | 91.249 ± 0.220       | 83.269 ± 0.449       |
> | StrDiffusion     | 34.255 ± 0.159     | 25.200 ± 0.236     | 49.560 ± 0.394       | 40.867 ± 0.336       |
> | HD-Painter       | 85.016 ± 0.626 | 69.089 ± 0.628 | 92.361 ± 0.026       | 87.389 ± 0.325       |
> | ASUKA            | 75.140 ± 1.022     | 58.686 ± 0.477     | 84.597 ± 0.458       | 73.614 ± 0.980       |
> | BrushNet         | 79.391 ± 0.375     | 57.046 ± 1.151     | 89.930 ± 0.396       | 85.064 ± 0.450       |
> | **BruPA (ours)** | 83.054 ± 0.886     | 61.081 ± 0.764     | 92.255 ± 0.163       | 88.686 ± 0.248 |
> | FLUX.1 Fill      | 83.935 ± 0.414     | 67.465 ± 0.463     | 92.752 ± 0.145 | 88.649 ± 0.306       |
> | **FluPA (ours)** | **87.609 ± 0.291** | **72.307 ± 0.560** | **94.501 ± 0.154**   | **91.603 ± 0.220**   |

---

> ### Author Response · Authors · 2025-11-23
>
> > **Weakness 3:** Quantitative results of biases
>
> Thank you for the suggestion. Because no established models exist specifically for brightness, color scheme, or composition, we make an attempt to employ **GPT-4** and **Qwen3-VL** for quantitative evaluation based on your advice. We report the system prompt and results in **Appendix D.1**, and we also put the **Table 9** below. The results demonstrate that HPSv2 consistently achieves the highest scores across brightness, composition, and color, whereas PickScore achieves the lowest scores across these dimensions. These quantitative findings align well with our qualitative observations in Section 6 that HPSv2 favors images with bright lighting, complex composition, and vivid colors; PickScore shows the opposite tendency; and our Ensemble falls squarely between the two
>
> **Table 9: Quantitative results of biases evaluated by GPT-4 and Qwen3-VL.**
>
> ### (a) GPT-4
>
> | model        | reward model | brightness     | composition    | color          |
> |-------------|--------------|----------------|----------------|----------------|
> | BrushNet    | HPSv2        | **79.221**     | **82.979**     | **81.267**     |
> | BrushNet    | Ensemble     | 78.180  | 80.922  | 78.513 |
> | BrushNet    | PickScore    | 73.545         | 75.019         | 70.641         |
> | FLUX.1 Fill | HPSv2        | **76.617**     | **74.547**     | **70.978**     |
> | FLUX.1 Fill | Ensemble     | 76.412  | 74.057  | 70.496 |
> | FLUX.1 Fill | PickScore    | 76.124         | 73.512         | 69.824         |
>
> ### (b) Qwen3-VL
>
> | model        | reward model | brightness     | composition    | color          |
> |-------------|--------------|----------------|----------------|----------------|
> | BrushNet    | HPSv2        | **83.843**     | **88.310**     | **85.565**     |
> | BrushNet    | Ensemble     | 83.284  | 88.180  | 84.644  |
> | BrushNet    | PickScore    | 82.615         | 86.530         | 82.516         |
> | FLUX.1 Fill | HPSv2        | **82.750**     | **85.832**     | **81.695**     |
> | FLUX.1 Fill | Ensemble     | 82.527  | 85.611  | 81.454  |
> | FLUX.1 Fill | PickScore    | 82.508         | 85.481         | 81.389         |
>
>
> > **Question 1:** Other ensemble strategies
>
> Thank you for the question. We have evaluated additional ensemble strategies. Please see our response to **Weakness 1**.

---

### Author Response · Authors · 2025-11-23
**General Response**

We sincerely thank all reviewers for their valuable comments and suggestions. We have carefully revised the manuscript, and all changes are highlighted in **blue**. As adding the new results directly into the main text would disrupt the layout, we have temporarily placed these additional results and discussions in the **Appendix**. In the final version, we will integrate them appropriately into the main text.

---

### Meta-Review · Area_Chair_3anD · 2025-12-17

**Summary:**

The reviewer acknowledged the contributions of the system explorations of ensemble reward methods and its strong test performance.
1. Nine reward models, multiple benchmark tests, and solid experimental results

2. This paper offers a detailed examination of the causes behind reward hacking. This analysis provides useful insights for the community.

3. The study explores how preference data for DPO can be constructed with respect to both sample scale and candidate scale, offering practical insights for future work in this area.

But reviewers expressed concerns primarily in the following aspects:

1. The proposed method is a simple ensemble strategy and lacks innovation.

2. Quantitative evaluations only relied on GPT4, leading to unreliable results. A more reliable assessment model should be validated.

3. The experimental setup lacks transparency; more implementation details should be included.

4. There is still no direct comparison with an another ensemble method proposed by Lee et al.

**Reviewer Concerns:**

The concerns raised by the reviewers mainly focus on the following aspects:
Cons.1: The proposed method is a simple ensemble strategy and lacks innovation. (Rev. mUgh, Rev. 9M76)

Res.1: The authors provided detailed clarifications, emphasizing that simplicity and effectiveness were their strategy.

Cons. 2: Quantitative evaluations that relied solely on GPT4 led to unreliable results. More reliable assessment models should be validated. ( Rev. mUgh, Rev. 9M76)

Res. 2:  The author added VLM Qwen3 as an additional evaluator and provided the quantitative results in Table 6.  Ten metrics were employed for evaluation as shown in Table 3 and Table 4.

Cons. 3: The experimental setup lacks transparency, with many details omitted. Although the authors emphasize scalability and effectiveness, the reproducibility is limited. (Rev. mUgh, Rev. XjVN, Rev. 9M76)

Res. 3  The authors added a detailed definition and algorithmic description of their implementation details in Appendix C.1.

Cons. 4: The authors claim that factors such as brightness, color, and composition can introduce bias into the inpainting model, but they have not provided quantitative analysis results or further investigated the underlying causes. (Rev. mUgh, Rev. XjVN, Rev. 9M76) Furthermore, their emphasis on the importance of these factors seems to ignore the crucial role of inpainting-text alignment, which is essential for many inpainting tasks (image restoration, object replacement). (Rev. XjVN)

Res. 4 The authors used VLM to perform a simple quantitative analysis of the performance of HPSv2 and PickScore in terms of brightness, composition, and color. They also evaluated the text alignment results using VLM on tasks such as image completion and object replacement, and reported the results in Table 12.

Cons. 5: Ensemble strategies need to incorporate more combination approaches, such as weighted combinations, learning combinations, or removing poorly performing reward models. (Rev. mUgh, Rev. XjVN, Rev. 9M76)

Res. 5  The authors added linear weighted ensemble and softmax weighted ensemble to conduct ensemble experiments. The results are listed in Table 7.

Cons. 6: There is no direct comparison with the method proposed by Lee et al. (Calibrated Multi-Preference Optimization for Aligning Diffusion Models), which is another ensemble method. (Rev. 854V)
Res. 6  The author does not conduct the comparison experiment due to time limitations. As a result, no comparison is provided; however, one is needed.

Overall, concerns 2-5 have been resolved, but 1 and 6 still require exploration.

**Reviewer Scores:**

The initial reviews were split into two parts. Two reviewers recommended acceptance, assigning scores of 8 (Accept) and 6 (Weak Accept), and noted the paper’s comprehensive empirical analysis. In contrast, the other two reviewers gave scores of 4 (Weak Reject), and concerns about technical novelty and evaluation metrics. During the rebuttal phase, the authors conducted additional ablation studies, including evaluating weighted ensemble strategies and removing low-quality reward models, and introduced new quantitative visual metrics and expert evaluations to strengthen the assessment of their method. Based on the review process and the rebuttal response, the main experimental concerns have been addressed. Overall, the paper’s strengths make up for its limitations.

---

### Decision · Program_Chairs · 2026-01-26

Accept (Poster)